# Modulation of tonotopic ventral medial geniculate body is behaviorally relevant for speech recognition

Paul Glad Mihai[1,2]*, Michelle Moerel[3,4,5], Federico de Martino[3,4,6], Robert Trampel[1], Stefan Kiebel[2], Katharina von Kriegstein[1,2]*

[1]Max Planck Institute for Human Cognitive and Brain Sciences, Leipzig, Germany; [2]Chair of Cognitive and Clinical Neuroscience, Faculty of Psychology, Technische Universität Dresden, Dresden, Germany; [3]Department of Cognitive Neuroscience, Faculty of Psychology and Neuroscience, Maastricht University, Maastricht, Netherlands; [4]Maastricht Brain Imaging Center (MBIC), Maastricht, Netherlands; [5]Maastricht Centre for Systems Biology (MaCSBio), Maastricht University, Maastricht, Netherlands; [6]Center for Magnetic Resonance Research, University of Minnesota, Minneapolis, United States

**Abstract** Sensory thalami are central sensory pathway stations for information processing. Their role for human cognition and perception, however, remains unclear. Recent evidence suggests an involvement of the sensory thalami in speech recognition. In particular, the auditory thalamus (medial geniculate body, MGB) response is modulated by speech recognition tasks and the amount of this task-dependent modulation is associated with speech recognition abilities. Here, we tested the specific hypothesis that this behaviorally relevant modulation is present in the MGB subsection that corresponds to the primary auditory pathway (i.e., the ventral MGB [vMGB]). We used ultra-high field 7T fMRI to identify the vMGB, and found a significant positive correlation between the amount of task-dependent modulation and the speech recognition performance across participants within left vMGB, but not within the other MGB subsections. These results imply that modulation of thalamic driving input to the auditory cortex facilitates speech recognition.
DOI: https://doi.org/10.7554/eLife.44837.001

**\*For correspondence:**
glad@posteo.de (PGM);
katharina.von_kriegstein@tu-dresden.de (KK)

**Competing interests:** The authors declare that no competing interests exist.

## Introduction

Human communication relies on fast and accurate decoding of speech—the most important tool available to us for exchanging information. Understanding the neural decoding mechanisms for speech recognition is important for understanding human brain function (*Rauschecker and Scott, 2009*), but also for understanding communication disorders such as developmental dyslexia (*Galaburda et al., 1994*; *Müller-Axt et al., 2017*). Since the early findings of *Wernicke (1874)* neuroscientific models of speech recognition have mainly focused on cerebral cortex mechanisms (*Hickok and Poeppel, 2007*; *Friederici and Gierhan, 2013*). Yet, more recently it has been suggested that a full understanding of speech recognition mechanisms might need to take the subcortical sensory pathways—particularly the sensory thalami—into account (*von Kriegstein et al., 2008a*; *Díaz et al., 2012*; *Díaz et al., 2018*; *Chandrasekaran et al., 2009*; *Chandrasekaran et al., 2012*).

The classic view, that the sensory thalamus is a passive relay station has been by-and-large abandoned over the last two decades. First of all, it is well known that there are strong corticofugal projections to the sensory thalamus (*Sherman and Guillery, 2006*; *Winer and Prieto, 2001*; *Lee and Sherman, 2012*; *Lee and Winer, 2011*). Furthermore, experimental evidence in humans and other

mammals in the visual as well as the auditory modality has shown that sensory thalamus responses are modulated by attention (*Saalmann and Kastner, 2011*), percept (*Haynes et al., 2005*), context (*Antunes and Malmierca, 2011*; *McAlonan et al., 2008*; *O'Connor et al., 2002*), and task (*Díaz et al., 2012*; *von Kriegstein et al., 2008b*; *Díaz et al., 2018*). Based on these findings, the sensory thalamus has become accepted as a structure that is modulated by cognitive demands and is more involved in active information regulation (*Saalmann and Kastner, 2011*; *Haynes et al., 2005*; *Antunes and Malmierca, 2011*; *McAlonan et al., 2008*; *O'Connor et al., 2002*; *Díaz et al., 2012*; *von Kriegstein et al., 2008b*), for a different take see *Camarillo et al. (2012)*.

In the case of speech, previous studies showed a task-dependent modulation in the auditory sensory thalamus for auditory speech recognition, (MGB; *von Kriegstein et al., 2008b*; *Díaz et al., 2012*) as well as a task-dependent modulation in the visual sensory thalamus for visual speech recognition (LGN; *Díaz et al., 2018*). In the MGB there were two findings: First, bilateral MGB showed significantly higher responses to an auditory speech recognition task than to control tasks, independent of attentional load or task-difficulty (*von Kriegstein et al., 2008b*; *Díaz et al., 2012*). Second, the performance level in the auditory speech recognition task was significantly correlated with the task-dependent modulation in the MGB of the left hemisphere across participants (*von Kriegstein et al., 2008b*).

Following the Bayesian brain hypothesis, (*Knill and Pouget, 2004*; *Friston and Kiebel, 2009*; *Friston, 2005*; *Kiebel et al., 2008*) and based on findings in non-human animals (*Krupa et al., 1999*; *Sillito et al., 1994*; *Wang et al., 2019*), one possible explanation for the MGB task-dependent modulation for speech is that cerebral cortex areas tune the sensory thalamus depending on behavioral demand, and that this tuning is particularly relevant for fast-varying and predictable stimuli such as speech (*von Kriegstein et al., 2008b*; *Díaz et al., 2012*). This view entails that the task-dependent modulation occurs already in those parts of the MGB that drive the cerebral cortex representations (*von Kriegstein et al., 2008b*)—the so-called first-order sensory thalamus (*Sherman and Guillery, 1998*).

The MGB consists of three divisions. Only the ventral MGB (vMGB) can be considered first-order sensory thalamus (*Malmierca et al., 2015* [review], *Winer et al., 2005* [review]), as vMGB receives driving inputs from sources that relay information from the sensory periphery and projects this information to the cerebral cortex (*Sherman and Guillery, 1998*). Ventral MGB also receives modulatory input from cerebral cortex (*Sherman and Guillery, 1998*). In contrast, the other two MGB divisions, the dorsal (dMGB) and medial MGB (mMGB), do not show major projections to primary auditory cortices (*Vasquez-Lopez et al., 2017*; *Anderson et al., 2007*; *de la Mothe et al., 2006*), and are not considered to be part of the first order (i.e., lemniscal) auditory pathway (*Anderson et al., 2007*; *Anderson et al., 2009*; *Calford, 1983*; *Cruikshank et al., 2001*; *Gonzalez-Lima and Cada, 1994*; *Hackett et al., 1998*; *Morest, 1964*; *Winer et al., 1999*); although see *Anderson and Linden (2011)*.

The goal of the present study was to test whether the behaviorally relevant task-dependent modulation for speech is located in the first-order auditory thalamus; that is, the vMGB (*von Kriegstein et al., 2008b*). Localization of the behaviorally relevant task-dependent modulation for speech to the vMGB would provide a crucial step forward in understanding sensory thalamus function for human cognition in vivo, as it would imply that the stimulus representation in the auditory sensory pathway is modulated when humans recognize speech.

Due to the relatively small size of human MGB (ca. 5 × 4 × 5 mm, *Winer, 1984*) and the spatial limitations of the non-invasive imaging techniques used in previous studies (*von Kriegstein et al., 2008b*; *Díaz et al., 2012*), it was so far not possible to differentiate in which of the three major MGB divisions there is behaviorally relevant task-dependent modulation for speech. Here we, therefore, used ultra-high field functional magnetic resonance imaging (fMRI) at 7 Tesla, enabling high spatial resolution measurements (*Duyn, 2012*). The vMGB has a strong tonotopic organization (*Calford, 1983*; *Rodrigues-Dagaeff et al., 1989*; *Anderson et al., 2007*) while the other two MGB subsections have only a weak tonotopic organization (i.e., broadly tuned neurons; *Anderson and Linden, 2011*; *Calford, 1983*; *Bartlett and Wang, 2011*; *Rodrigues-Dagaeff et al., 1989*; *Ohga et al., 2018*). We planned to distinguish the vMGB based on its tonotopic organization as well as its topographic (i.e., ventral) location.

We employed three fMRI paradigms—an MGB localizer, a tonotopy localizer, and the speech experiment. In the MGB localizer and the tonotopy localizer (*Figure 1A*), participants listened to

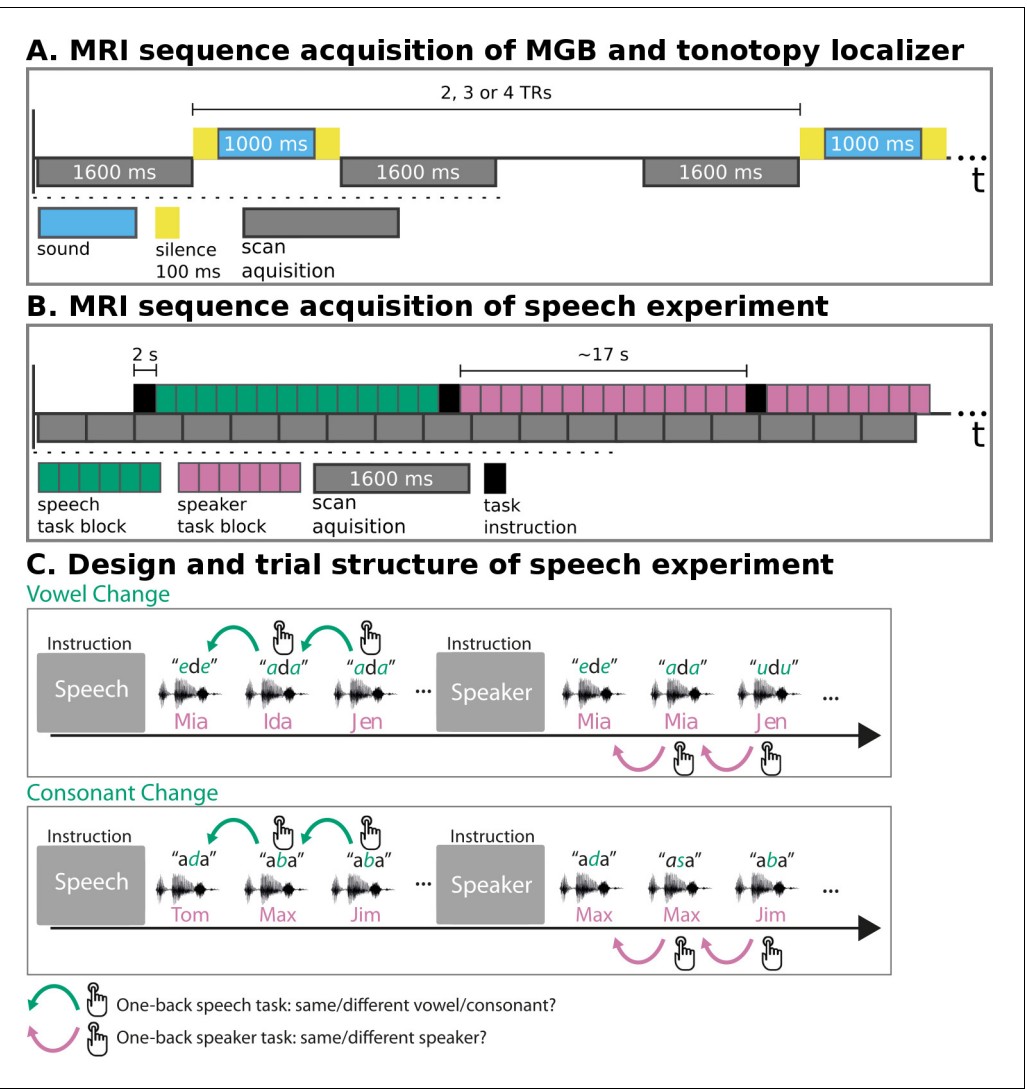

**Figure 1.** MRI sequence acquisition and experimental design. (**A**) MRI sequence acquisition of MGB and tonotopy localizer. Stimuli ('sound') were presented in silence periods between scan acquisitions and jittered with 2, 3, or 4 TRs. TR: repetition time of volume acquisition. (**B**) MRI sequence acquisition of the speech experiment. Each green or magenta rectangle of a block symbolizes a syllable presentation. Blocks had an average length of 17 s. Task instructions ('speech', 'speaker') were presented for 2 s before each block. MRI data were acquired continuously ('scan acquisition') with a TR of 1600 ms. (**C**) Design and trial structure of speech experiment. In the speech task, listeners performed a one-back syllable task. They pressed a button whenever there was a change in syllable in contrast to the immediately preceding one, independent of speaker change. The speaker task used exactly the same stimulus material and trial structure. The task was, however, to press a button when there was a change in speaker identity in contrast to the immediately preceding one, independent of syllable change. Syllables differed either in vowels or in consonants within one block of trials. An initial task instruction screen informed participants about which task to perform.

DOI: https://doi.org/10.7554/eLife.44837.002

natural sounds (human voices, animal cries, tool sounds) (*Moerel et al., 2015*). The MGB localizer served to identify the left and right MGB. The tonotopic maps resulting from the tonotopy localizer were used to localize the left and right vMGB. These served as regions of interest for hypothesis testing in the speech experiment. In the speech experiment (*Figure 1B and C*), participants listened to blocks of auditory syllables (e.g.,/aba/), and performed either a speech or a speaker task. In the speech task, participants reported via button press whether the current syllable was different from

the previous one (1-back task). In the speaker task, participants reported via button press whether the current speaker was different from the previous one.

In previous studies we found the task-dependent modulation for speech (i.e., higher response in the speech in contrast to a control task on the same stimulus material) in both the left and right MGB and a correlation of the task-dependent modulation with speech recognition performance only in the left MGB (*von Kriegstein et al., 2008a*; *Díaz et al., 2012*). Since our aim of the present paper was to test whether the behaviourally-relevant task dependent modulation of MGB is present in the vMGB, we tested two hypotheses. We hypothesized (i) a higher response to the speech than to the control (speaker) task in the tonotopically organized left and right vMGB, and (ii) a positive correlation between speech recognition performance and the task-dependent modulation for speech in the tonotopically organized left vMGB. Within our design these hypotheses could be addressed by (i) the main effect of task (speech task vs speaker task) in bilateral vMGB and by (ii) a correlation between the contrast speech task vs speaker task with speech recognition performance across participants in left vMGB.

## Results

### Tonotopy localizer – replication of tonotopy in MGB

First, we replicated the MGB tonotopy reported previously by *Moerel et al. (2015)* with a larger participant sample. Participants listened to natural sounds (human voices, animal cries, tool sounds) in a fast event-related scheme during silent gaps of the clustered imaging technique (*Moerel et al., 2015*) (*Figure 1A*). Using a model that mimics peripheral sound processing (*Chi et al., 2005*), each sound was represented as a spectrogram. The resulting spectrograms were averaged over time and divided into ten equal bandwidths in octaves. Onsets for each bin were convolved with the hemodynamic response function and entered into the general linear model. Each voxel within each participant's left and right MGB localizer mask was labeled according to the frequency bin to which it responded strongest, that is which had the highest parameter estimate (*Moerel et al., 2015*). Thus, voxels would have values from 1 to 10 corresponding to the frequency bin that they best represented. This resulted in a map of frequency distributions from low to high frequencies in the left and right MGB for each participant.

Similar as in *Moerel et al. (2015)*, we found two tonotopic gradients within the MGB in the group analysis. On visual inspection, one high frequency region in the middle of the MGB was flanked by gradually lower frequency components dorsally and ventrally (*Figure 2A and B*). The two gradients run dorso-lateral to ventro-medial.

The regions of low and high frequency preference could be observed in the sagittal view. To quantify the tonotopic gradient direction, we calculated gradient angles in ten slices of the left and right tonotopic map in sagittal orientation. Histograms of gradient angles in 5° steps were calculated for each slice. The histograms of the gradients were then averaged first over slices per participant, followed by an average over participants. The analysis of the mean gradient distributions across individuals (*Figure 3*, black line with standard error of the mean in gray) for the left MGB had maxima at 130° and 300° (dashed red lines, *Figure 3*). In the right MGB the mean across individual distributions had maxima at 130° and 310°.

For completeness we ran the same tonotopy analysis also on the inferior colliculi (IC). This analysis (n = 28) revealed a single gradient in the IC similarly to previous reports in the macaque (n = 3) (*Baumann et al., 2011*) and human (n = 6; n = 5) (*Moerel et al., 2015*; *De Martino et al., 2013*) (*Figure 3—figure supplement 1*).

### Tonotopy localizer—Localisation of vMGB

We used the high frequency components in the middle of the MGB as a reference to subdivide the MGB volume into two regions per hemisphere (*Figure 4A and B*). For the left MGB, gradient one was located ventrally and slightly medial compared to gradient 2, which was situated more anterior, dorsal, and lateral. For the right MGB we found similar locations: gradient one was more ventral and medial compared to gradient 2. The center of mass (COM) and the volume for each region is summarized in *Table 1*. Based on the tonotopy and its ventral location (*Morel et al., 1997*; *Bartlett and Wang, 2011*) we considered gradient one to represent the vMGB (*Moerel et al., 2015*).

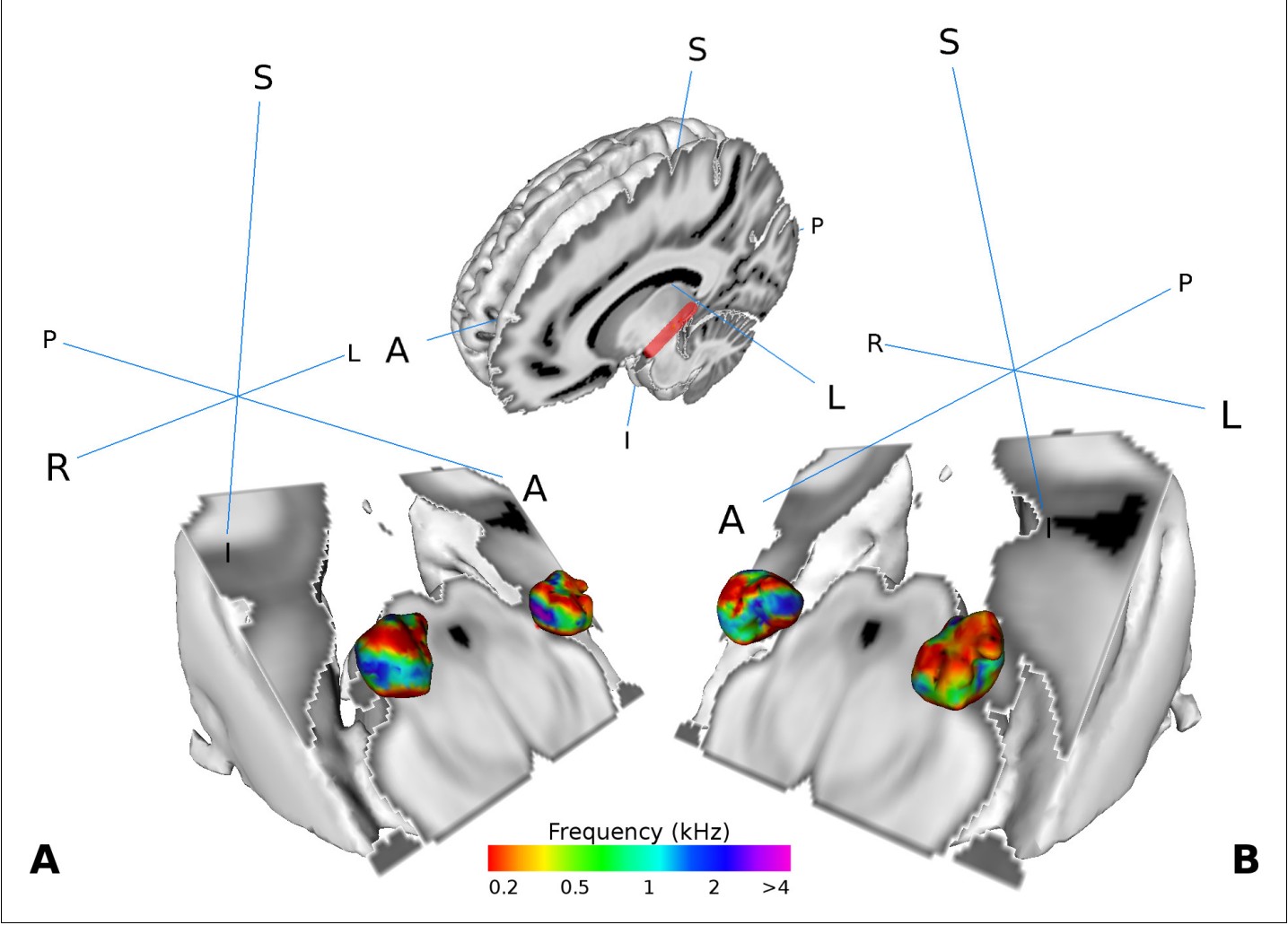

**Figure 2.** Visualization of the average tonotopy across participants (n = 28) found in the MGB using the tonotopic localizer. The half-brain image at the top shows the cut through the brain with a red line denoting the −45° oblique plane used in the visualizations in panels A-B. (A) Three dimensional representation of the tonotopy in the left and right MGB with two low-high frequency gradients. (B) Same as in A with a different orientation. Crosshairs denote orientation (A: anterior, P: posterior, L: left, R: right, S: superior, I: inferior).

DOI: https://doi.org/10.7554/eLife.44837.003

## Speech experiment

### Behavioral results

Participants scored a mean hit rate in the speech task of 0.872 with 97% highest posterior density (HPD) interval [0.828, 0.915], and a mean hit rate in the speaker task of 0.760 with a 97% HPD interval [0.706, 0.810] (*Figure 5*; *Figure 5—figure supplement 1*). The mean hit-rate was 0.112 higher in the speech task than in the speaker task with 97% HPD interval [0.760, 0.150].

An analysis of reaction times showed that there was no speed-accuracy trade-off for the two tasks (*Figure 5—figure supplement 2*).

### fMRI results

Using the fMRI data of the speech experiment we tested our two hypotheses. The first hypothesis was that within the ventral tonotopic gradient (i.e., vMGB) there is a task-dependent modulation (i.e., higher responses for the speech than the speaker task). Unexpectedly, there was no evidence for higher BOLD response in the speech task in comparison to the speaker task (speech vs speaker contrast) in vMGB.

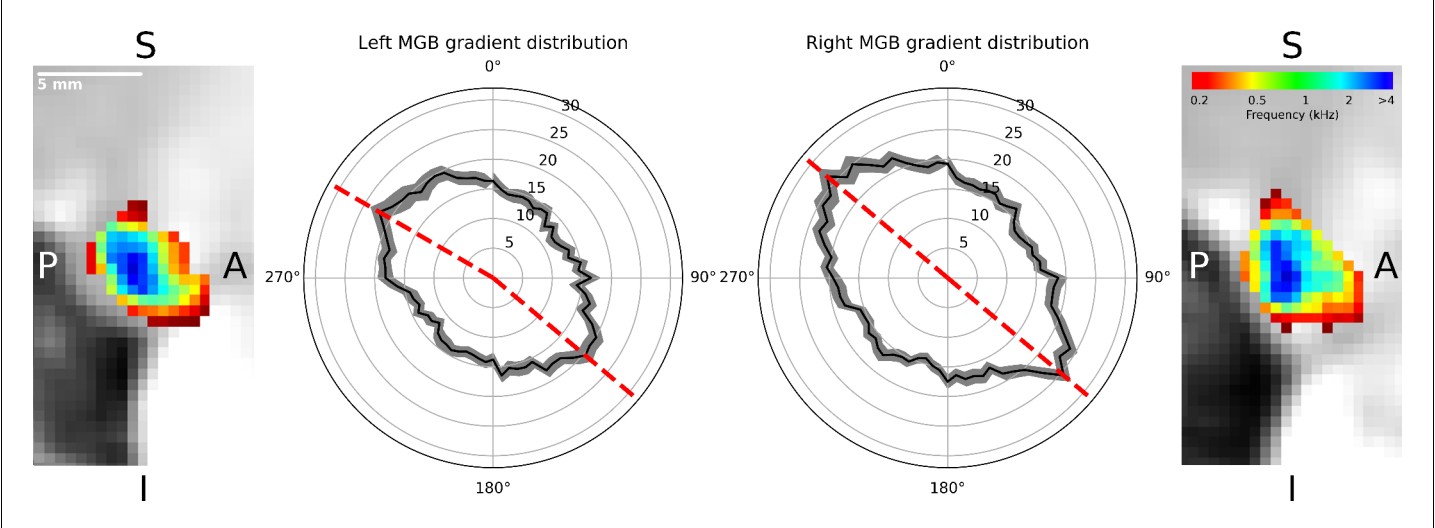

**Figure 3.** Distribution of gradients in a sagittal plane for ten slices averaged over participants (n = 28). The mean number of angle counts in 5° steps (black line with standard error of the mean in gray, numbers indicate counts) for the left MGB have maxima at 130° and 300° (red dashed lines). For the right MGB the maximum gradients are at 130° and 310° (red dashed lines). We interpreted these as two gradients in each MGB: one from anterior-ventral to the center (130°) and the other from the center to anterior-dorsal-lateral (300°, 310°). The two outer images display a slice of the mean tonotopic map in the left and right MGB in sagittal view (S: superior, I: inferior, P: posterior, A: anterior).

DOI: https://doi.org/10.7554/eLife.44837.004

The following figure supplement is available for figure 3:

**Figure supplement 1.** Mean distribution of gradients for the inferior colliculus (IC) in coronal view for three slices averaged over participants (n = 28).
DOI: https://doi.org/10.7554/eLife.44837.005

Our second hypothesis was that there is a positive correlation between speech recognition performance and the amount of task-dependent modulation for speech (i.e., speech vs speaker contrast) in the left vMGB across participants. As expected, there was a significant correlation between the speech vs speaker contrast and mean percent correct speech recognition scores across participants in the left vMGB [MNI coordinate:$(-11, -28, -5)$; SVC for vMGB $p = 0.04$ FWE, $T = 2.97$, $r = 0.46$ using T to r transform from *Fisher (1915)*; parameter estimate (β) and 90% CI 0.82 [0.36, 1.27]; *Figures 6* and *7*]. The correlation coefficient r=0.46 ($R^2$=0.21) is considered to represent a large effect (*Cohen, 1988* p. 80), explaining 21% of the variance of either variable when linearly associated with the variance in the other.

To check whether potential outliers were driving the correlation, we excluded those data points that were two standard deviations away from the parameter estimate mean. One data point was outside this threshold. The re-calculated correlation was very similar to the one with all data points ($T = 2.92$, $p = 0.038$, $r = 0.46$), indicating that the correlation was robust to outlier removal.

## Meta-analysis of the main effect of task (speech vs speaker contrast)

We performed a random effects meta-analysis to test whether the (non-significant) effect of the main effect of task in the present study (i.e., speech vs speaker task contrast) was different from other studies that have reported a significant task-dependent MGB modulation for speech. We included five studies in the meta-analysis that each contained a speech task vs control task contrast: two experiments from *von Kriegstein et al. (2008b)*, the data from the control participants of *Díaz et al. (2012)*, the result of a recent study (*Mihai et al., 2019*), and the result of the current study. The meta-analysis yielded an overall large effect size of d = 0.85 [0.06, 1.65], p=0.036 (*Figure 8*). The analysis showed that four of the five experiments had a positive medium to large effect and only the current study had a very small insignificant negative effect. The confidence intervals from the current study also do not overlap with those of the other studies. In terms of equivalence testing (*Schuirmann, 1987*), this means that the result of the speech task vs control task contrast of

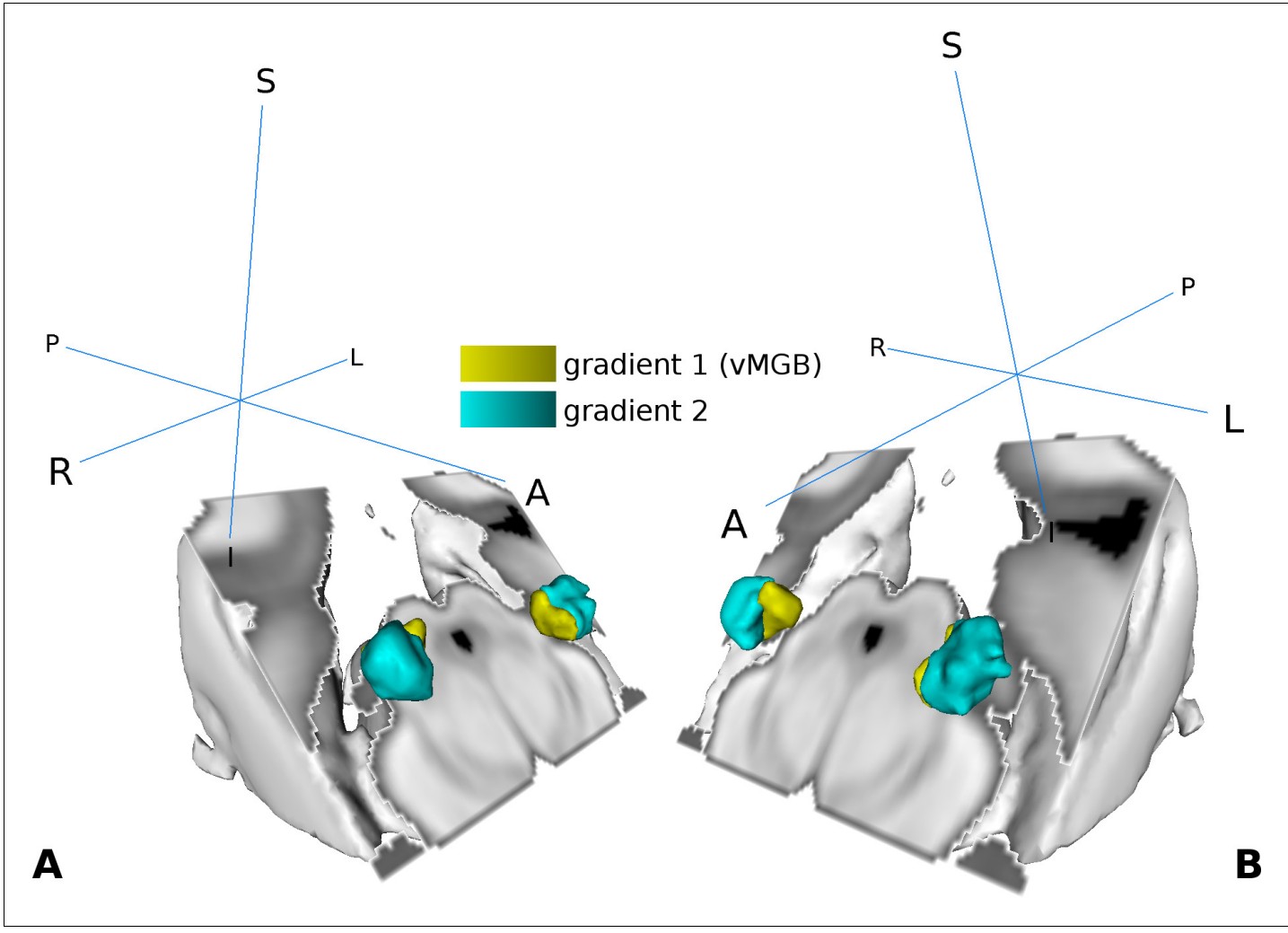

**Figure 4.** Visualization of the tonotopic gradients found in the MGB based on the tonotopic localizer (see *Figure 2*). (A) Three dimensional rendering of the two tonotopic gradients (yellow: ventro-medial gradient 1, interpreted as vMGB, cyan: dorso-lateral gradient 2) in the left and right MGB. (B) Same as in A with a different orientation. Orientation is the same as in *Figure 2*; crosshairs denote orientation.

DOI: https://doi.org/10.7554/eLife.44837.006

the current study is different from the other studies compared here. We detail potential reasons for this difference in the task-dependent modulation between the studies in the discussion.

## Exploratory analyses

*Specificity of the behaviorally relevant task-dependent modulation for speech.* In exploratory control analyses we checked whether we could test for a specificity of the correlation between the task-dependent modulation for speech (i.e., the speech task vs speaker task contrast) and the speech

**Table 1.** Center of mass (COM) and volume of each MGB mask used in the analysis.

| Mask | COM (MNI coordinates mm) | Volume (mm³) |
|---|---|---|
| Left Gradient 1 (ventro-medial) | (−12.7,–26.9, −6.3) | 37.38 |
| Left Gradient 2 (dorso-lateral) | (−14.8,–25.9, −5.4) | 77.38 |
| Right Gradient 1 (ventro-medial) | (12.7,–27.6, −4.4) | 45.00 |
| Right Gradient 2 (dorso-lateral) | (14.7,–25.8, −4.3) | 67.38 |

DOI: https://doi.org/10.7554/eLife.44837.007

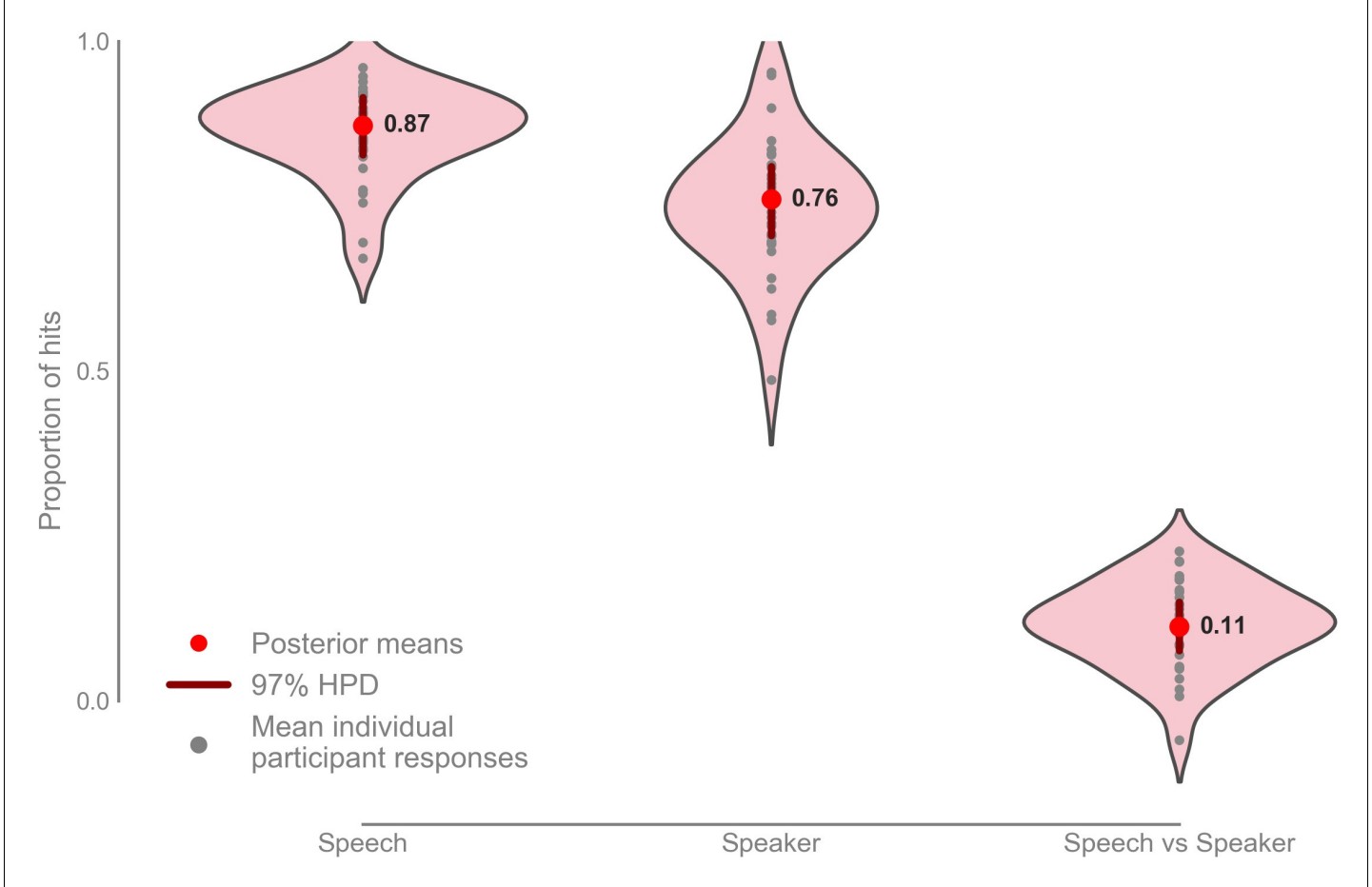

**Figure 5.** Mean proportion of correct button presses for the speech and speaker task behavioral scores, as well as the difference between the speech and speaker task (n = 33). Mean speech task: 0.872 with 97% HPD [0.828, 0.915], mean speaker task: 0.760 with 97% HPD [0.706, 0.800], mean speech vs speaker task: 0.112 with 97% HPD [0.760, 0.150]. Raw data provided in the Source Data File.

DOI: https://doi.org/10.7554/eLife.44837.008

The following source data and figure supplements are available for figure 5:

**Source data 1.** Raw behavioral correct button presses and total correct answers expected.
DOI: https://doi.org/10.7554/eLife.44837.011

**Figure supplement 1.** Mean proportion of correct button presses for each condition.
DOI: https://doi.org/10.7554/eLife.44837.009

**Figure supplement 2.** Results from the reaction time analysis.
DOI: https://doi.org/10.7554/eLife.44837.010

recognition behavior across participants. This was, however, not possible. Performance in the speech and the speaker tasks was significantly correlated ($r = 0.77$, p<0.001). Accordingly, there was no difference in the correlation between the contrast speech vs speaker task and the speech recognition scores and the correlation of the contrast speech vs speaker with the speaker recognition score ($z = -0.717$, p=0.474). The results indicated that although there is a behaviorally relevant task-dependent modulation in the vMGB for speech, we currently do not know whether it is specific to speech recognition abilities.

*Specificity of the behaviorally relevant task-dependent modulation to the vMGB.* In an exploratory analysis we checked whether the correlation between the task-dependent modulation for speech and the speech recognition behavior across participants was specific to the vMGB in contrast to other MGB subsection. There was no correlation (speech vs speaker task correlated with speech recognition performance across participants) in the left MGB–gradient 2 (*Supplementary file 1*). The comparison of the two correlations (i.e., for left vMGB and left MGB-gradient 2) showed that they

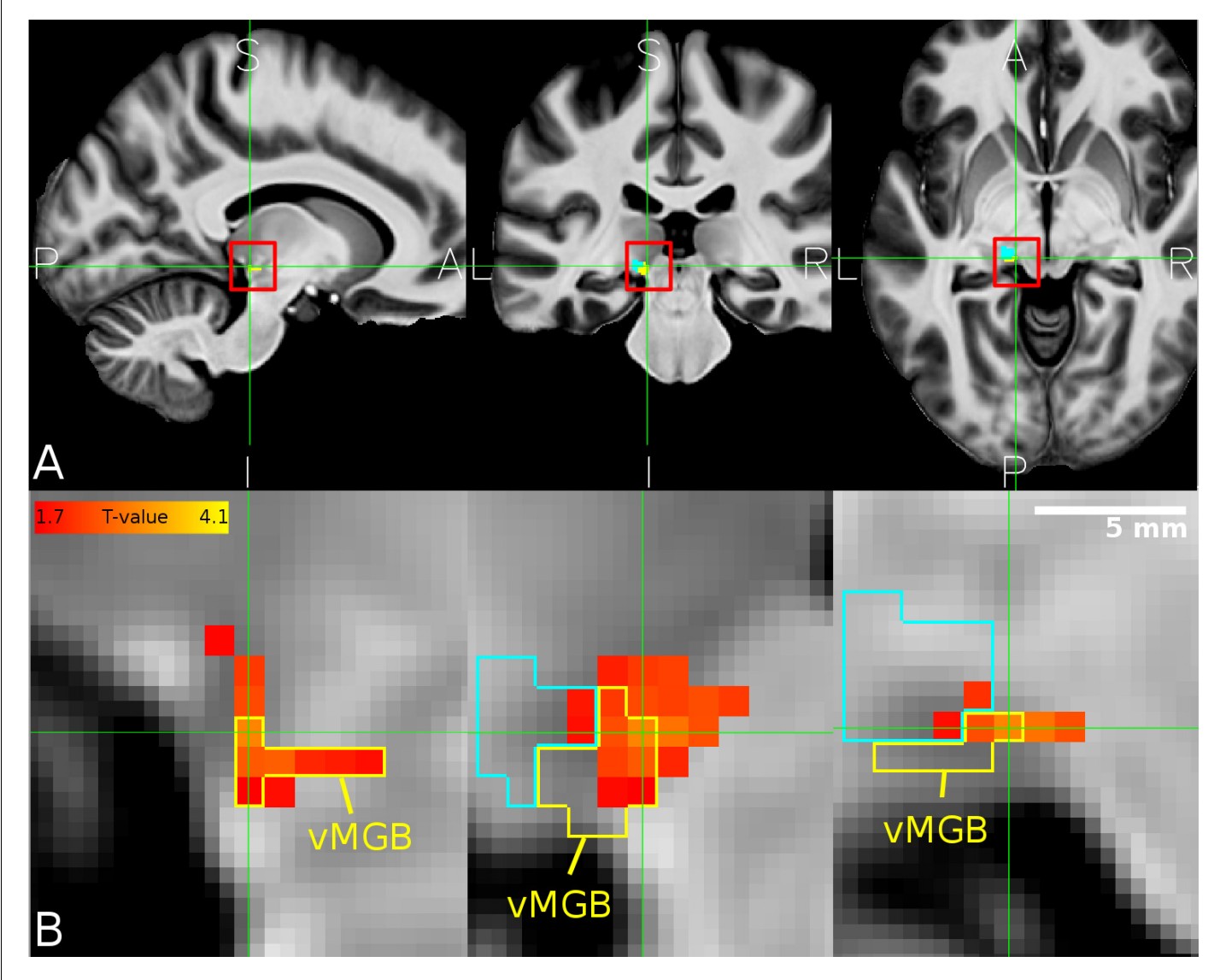

**Figure 6.** Overlap between MGB divisions and the behaviourally relevant task-dependent modulation. (**A**) The mean structural image across participants (n = 33) in MNI space. The red squares denote the approximate location of the left MGB and encompass the zoomed in view in B. (**B**) Overlap of correlation between the speech vs speaker contrast and the mean percent correct in the speech task (hot color code) across participants within the left vMGB (yellow). The tonotopic gradient two is shown in cyan. Panels correspond to sagittal, coronal, and axial slices (P: posterior, A: anterior, S: superior, I: inferior, L: left, R: right). Crosshairs point to the significant voxel using SVC in the vMGB mask (MNI coordinate −11,–28, −5).
DOI: https://doi.org/10.7554/eLife.44837.012

The following figure supplements are available for figure 6:

**Figure supplement 1.** Example of the orientation and volume covered by the 28 slices of the functional MRI measurements.
DOI: https://doi.org/10.7554/eLife.44837.013

**Figure supplement 2.** An example of an echo planar image in sagittal, axial and coronal view, as recorded in the speech experiment.
DOI: https://doi.org/10.7554/eLife.44837.014

**Figure supplement 3.** Example workflow for the functional MRI analysis of the speech experiment as coded in nipype depicting different processing stages (preprocessing, Register to MNI, First level design and estimation, overlay output for quick inspection).
DOI: https://doi.org/10.7554/eLife.44837.015

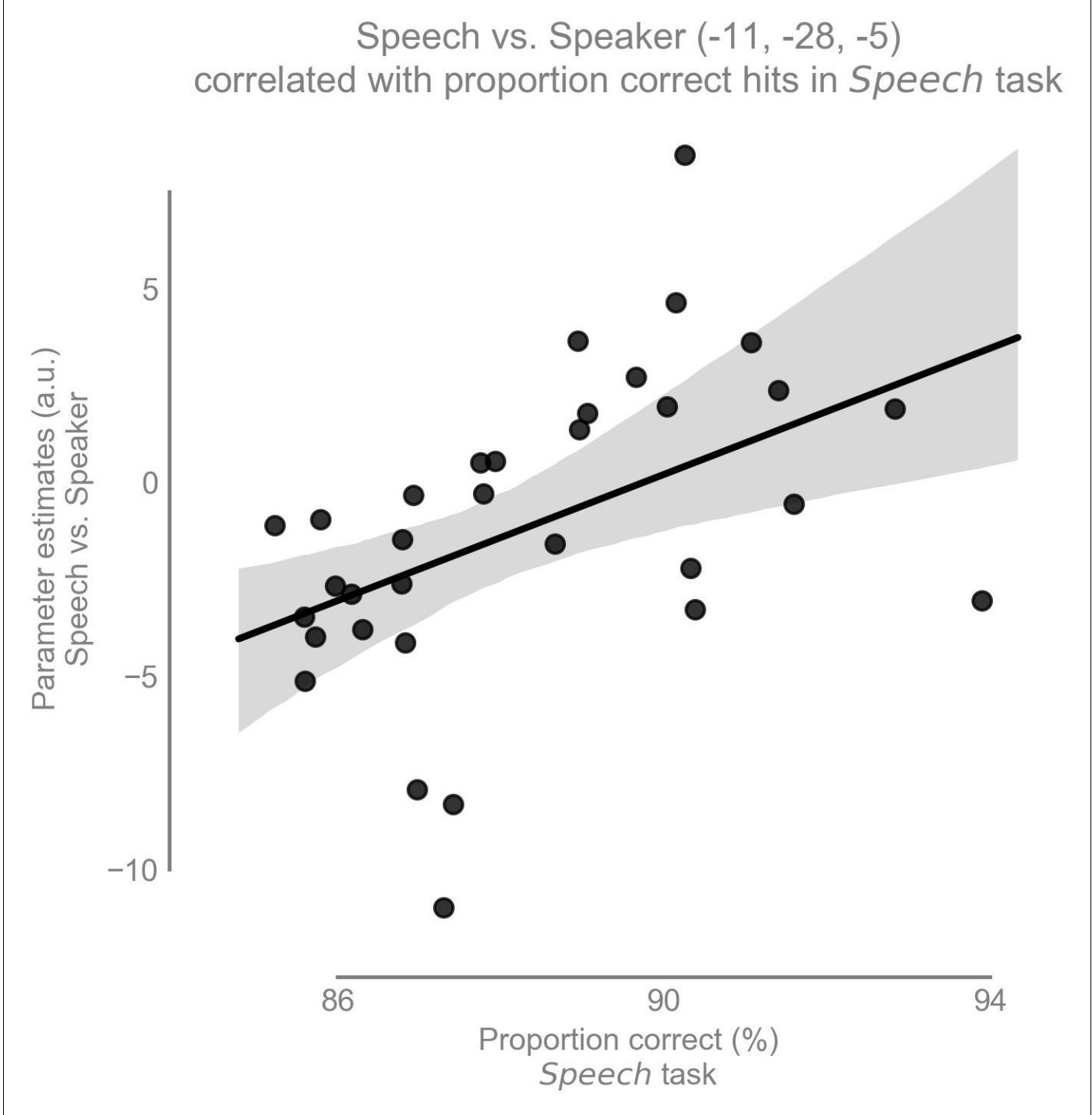

**Figure 7.** Task-dependent modulation of left vMGB correlates with proportion correct responses in the speech task over participants (n = 33): the better the behavioral score in the speech task, the stronger the BOLD response difference between speech and speaker task in the left vMGB (maximum statistic at MNI coordinate [−11,–28, −5]. The line represents the best fit with 97% bootstrapped confidence interval (gray shaded region).
DOI: https://doi.org/10.7554/eLife.44837.016

The following figure supplement is available for figure 7:

**Figure supplement 1.** Correlation of parameter estimates (Speech vs. Speaker) for the significant voxel in the vMGB in the speech vs. speaker task with percent correct behavioral score in the speech task.
DOI: https://doi.org/10.7554/eLife.44837.017

are significantly different (z = 4.466, p<0.0001). Thus, this is a first indication that the behaviorally relevant task-dependent modulation for speech is specific to the left vMGB.

*Exploration of other subcortical areas.* In exploratory analyses we tested the main effect of task (speech vs speaker contrast) and the correlation of the speech vs speaker contrast with behavioral performance in the speech task in several areas outside the vMGB, for which we did not have an a priori hypothesis. There were no significant effects in any other regions (*Supplementary file 1*).

## Discussion

Using ultra-high field fMRI we showed that it is the left auditory first-order sensory thalamus – the left ventral subdivision of the MGB (vMGB) – that is modulated during speech recognition. The vMGB is the primary sensory pathway nucleus of the auditory thalamus and transmits input to the cerebral cortex (*Winer, 1984*; *Anderson et al., 2007*; *Bartlett et al., 2011*; *Bordi and LeDoux, 1994*; *Calford, 1983*; *Moerel et al., 2015*). The present results imply that, when decoding speech, higher order cortical areas modify representations of the sensory input in the primary sensory thalamus and that such modification is relevant for speech recognition abilities. These results are a further indication that speech recognition might only be fully understood if dynamic cortico-thalamic interactions are taken into account (*Klostermann et al., 2013*; *von Kriegstein et al., 2008a*). The results are based on the test of two equally weighted hypotheses: a main effect and a correlation. Although we found no significant main effect in the vMGB (nor in any other subregion of the MGB), the correlation between the task-dependent modulation and behavioral performance was, as hypothesized, significant within the left vMGB.

We localized the vMGB based on its tonotopic organization and location relative to other MGB divisions. The tonotopic organization of the vMGB has been observed in many species with the use of invasive techniques (*Winer, 1984*; *Anderson et al., 2007*; *Bartlett et al., 2011*; *Bordi and LeDoux, 1994*; *Calford, 1983*) and non-invasively in six human participants using ultra-high field fMRI (*Moerel et al., 2015*). Similar to *Moerel et al. (2015)* we here also identified two tonotopic gradients. *Moerel et al. (2015)* attributed the ventral gradient to the ventral MGB, and the other gradient cautiously to the tonotopically organized lateral posterior thalamic nucleus (Pol), which is part of the non-lemniscal system (*Jones, 1985*). The Pol is also tonotopically organized with sharp tuning curves similar to the vMGB (*Imig and Morel, 1985*). Gradient two in our study is, however, larger than gradient 1. Thus, gradient two might also represent a composite of several nuclei that are in close proximity to the MGB (*Bartlett and Wang, 2011*) such as the Pol and potentially the suprageniculate, which has a preference for high frequencies (*Bordi and LeDoux, 1994*) (for a detailed thalamic atlas see *Morel et al., 1997*). Furthermore, the weak tonotopy of the dMGB or mMGB might also contribute to gradient 2. Another interpretation of the two tonotopic gradients is that the vMGB in humans might include two tonotopic maps, that is that frequency gradient 1 and 2 are part of the vMGB. Two tonotopic gradients have been found in the rat vMGB (*Shiramatsu et al., 2016*), but not consistently in other species (*Hackett et al., 2011*; *Horie et al., 2013*; *Tsukano et al., 2017*). The volume of the two gradients, however, speaks against the possibility of two tonotopic maps in human vMGB. That is, the two gradients make up already ca. 100 mm$^3$ and reported whole MGB volumes based on characterization in post-mortem human brains are between ca. 40–120 mm$^3$ (*Rademacher et al., 2002*; *Moro et al., 2015*). Thus, while gradient one can be clearly attributed to the vMGB, due to its tonotopic gradient and ventral location, the nature of the second frequency gradient remains an open question.

Based on previous findings (*von Kriegstein et al., 2008b*; *Díaz et al., 2012*), we expected significant responses for the speech vs speaker task contrast in vMGB. The lack of a significant main effect of task (speech vs speaker) in the vMGB was surprising, and the meta-analysis showed that the null-finding was indeed different from categorical task-effects (speech vs loudness tasks and speech vs speaker tasks) observed in other experiments in participants with typical development (*von Kriegstein et al., 2008b*; *Díaz et al., 2012*; *Mihai et al., 2019*).

There are three potential explanations for the difference in the results for the main effect of task between the present and previous studies. First, the speaker task was more difficult to perform (indicated by the lower behavioral score during the speaker vs speech task and subjective reports of the participants), which may have led to higher BOLD responses for the more difficult task. However, this explanation is unlikely as previous studies with matched performance across tasks (*Díaz et al.,*

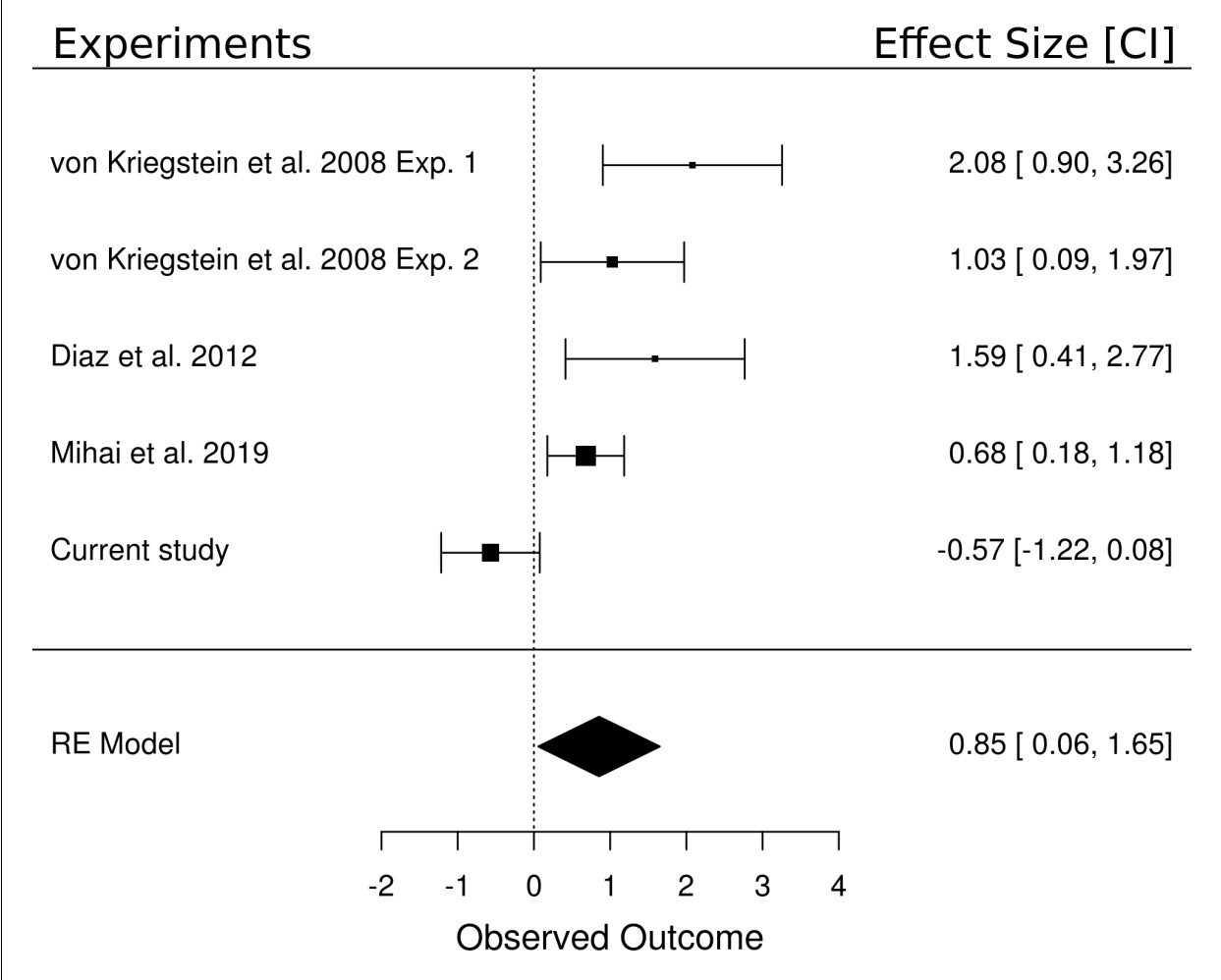

**Figure 8.** Meta-analysis of five experiments that investigated the difference between a speech and a non-speech control task. Experiment 1 of *von Kriegstein et al. (2008b)* tested a speech task vs loudness task contrast (n = 16). All other experiments included a speech task vs speaker task contrast (i.e., Experiment 2 of *von Kriegstein et al., 2008b* (n = 17), the control participants of *Díaz et al., 2012* (n = 14), the result of a recent experiment (*Mihai et al., 2019*) (n = 17) as well as the current study (n = 33)). The meta-analysis yielded an overall large effect size of d = 0.85 [0.06, 1.65], p=0.036. The area of the squares denoting the effect size is directly proportional to the weighting of the particular study when computing the meta-analytic overall score.

DOI: https://doi.org/10.7554/eLife.44837.018

*2012*; *von Kriegstein et al., 2008b* experiment 2) and also studies where the control task was more difficult than the speech task (*von Kriegstein et al., 2008b* experiment 1) have found a task-dependent MGB modulation for the speech task.

Second, we employed a liberal threshold in choosing participants based on their reading speed and comprehension scores (lower fourth of the mean and higher, i.e., 26–100%). Participants who scored lower on this test might also show a lower task-dependent modulation of the MGB (see *Díaz et al., 2018*). However, we find this explanation unlikely as those participants with lower reading score showed a broad (low to high) BOLD-response spectrum (*Figure 7—figure supplement 1*), and in the previous study (*Díaz et al., 2018*) a correlation between the MGB task-dependent modulation and reading speed and comprehension scores has been found only in participants with developmental dyslexia, but not in typically developed controls.

A third explanation is that we used unmanipulated natural voices from different speakers. In the previous studies different speaker voices were synthesized from one original voice to differ only in two key voice-identity parameters, that is the acoustic effect of the vocal tract length and the fundamental frequency (f0) (*von Kriegstein et al., 2008b*; *Díaz et al., 2012*; *Gaudrain et al., 2009*). Vocal

tract length and f0 are relatively stable acoustic cues that do not vary greatly over time in contrast to the highly dynamic cues (e.g., formant transitions, voice onset times, stops, *Kent et al., 1992*) that are most important for signaling phonemes and are used for speech recognition. However, dynamic cues, such as pitch periodicity, segmental timings, and prosody can also be used for speaker identification (*Benesty et al., 2007*). In the present experiment, which included natural voices, participants might have also used fast changing cues for speaker identity recognition, particularly because the task was difficult. Since dynamic cues are essential for speech recognition, using dynamic cues in a speaker task would render the two tasks less different. Thus, MGB modulation might also have played a role in performing the speaker task.

The localization of the correlation between the speech vs speaker contrast and performance in the speech task to the vMGB confirmed our hypothesis that it is the first-order thalamic nucleus (vMGB) that is involved in speech recognition. The correlation indicated that those participants on the lower side of the task-dependent modulation spectrum, as given by the speech vs speaker contrast, have lower proportion of hits in the speech task and those participants on the higher side of the task-dependent modulation spectrum, show higher proportion of hits in the speech task and are thus better at speech recognition. The variability of proportion of hits across participants was between 85.2 and 93.9% which translates to an odds ratio of 2.66 (i.e., those participants with higher task-dependent modulation are ~2.5 times more likely to have more hits compared to misses in the speech task). This variability is meaningful for a speech recognition task; for example, it has been shown that the hearing impaired perform 5–10% lower on speech recognition compared to normal hearing participants (*Panouillères and Möttönen, 2018*; *Gordon-Salant and Fitzgibbons, 1993*).

In the present study, the results cannot be explained by differences in stimulus input in the two conditions, as the same stimuli were heard in both tasks. We, however, cannot exclude with the present data set that, in general, participants who are better task-performers have a higher task-dependent modulation of the vMGB for speech. It is therefore still an open question whether the task-dependent modulation is specific to speech recognition behavior. A previous study in the visual modality, gave first evidence that the task-dependent modulation of the visual thalamus (LGN) for visual speech is specific for predictable dynamic information such as visual speech in contrast to unpredictable dynamic information (*Díaz et al., 2018*). However a similar study in the auditory modality is so far missing.

What kind of mechanism could be represented by the correlation between task-dependent modulation of the vMGB and speech recognition performance? Experimental and theoretical accounts of brain function emphasize the importance of an anatomical cortical and subcortical hierarchy that is organized according to the timescale of complex stimuli in the natural environment (*Davis and Johnsrude, 2007*; *Giraud et al., 2000*; *Kiebel et al., 2008*; *Wang et al., 2008*). In brief, it is assumed that levels closer to the sensory input encode faster dynamics of the stimulus than levels further away from the sensory input. In accordance with this view, the MGB (as well as the visual first-order thalamus [LGN]; *Hicks et al., 1983*) is tuned to high frequencies of temporal modulation (ca. 16 Hz in human MGB; *Giraud et al., 2000*) in relation to their associated primary sensory cortical areas (*Giraud et al., 2000*; *Wang et al., 2008*; *Foster et al., 1985*). For humans, the optimized encoding of relatively fast dynamics; for example at the phoneme level, is critical for speech recognition and communication (*Shannon et al., 1995*; *Tallal et al., 1996*; *Tallal and Piercy, 1975*). Many important speech components like formant transitions, voice onset times, or stops are on very fast time scales of 100 ms or less (*Hayward, 2000*). Additionally, the sound envelope described by relatively fast temporal modulations (1–10 Hz in quiet environments, 10–50 Hz in noisy environments) is important for speech recognition (*Elliott and Theunissen, 2009*; ; *Shannon et al., 1995*). The Bayesian brain hypothesis proposes that the brain uses internal dynamic models of its environment to predict the trajectory of the sensory input (*Knill and Pouget, 2004*; *Friston and Kiebel, 2009*; *Friston, 2005*; *Kiebel et al., 2008*). In accordance with this hypothesis, we have previously suggested that slower dynamics encoded by auditory cortical areas (*Giraud et al., 2000*; *Wang et al., 2008*) provide predictions about input arriving at lower levels of the temporal-anatomic hierarchy (*Kiebel et al., 2008*; *von Kriegstein et al., 2008b*). In this view, these dynamic predictions modulate the response properties of the first-order sensory thalamus to optimize the early stages of speech recognition. For example, in non-human animals cortico-thalamic projections outnumber thalamo-cortical projections (reviewed in *Ojima and Rouiller, 2011*), and alter the response properties of thalamic neurons (*Andolina et al., 2007*; *Cudeiro and Sillito, 2006*; *Ergenzinger et al., 1998*;

*Ghazanfar and Nicolelis, 2001*; *Sillito et al., 2006* [review]; *Wang et al., 2018*; *Sillito et al., 1994*; *Krupa et al., 1999*). In speech processing such a mechanism might be especially useful as the signal includes both rapid dynamics, and is predictable (e.g., due to co-articulation or learned statistical regularities in words *Saffran, 2003*). We suggest that higher-level regularities yield the predictions for lower-level details of the stimulus. While, for example, words are represented at the level of the cerebral cortex (*Huth et al., 2016*; *Davis and Johnsrude, 2003*; *Price et al., 2005*), the predictions about the most likely components of the word are percolating down the hierarchy. In this view, sensory thalamus structures do not represent word level information but they are tuned by the cerebral cortex areas to expect the detailed trajectory of the speech signal that is most likely given the expectations generated at the cerebral cortex level. Speech needs to be computed online often under suboptimal listening conditions. Building up accurate predictions within an internal generative model about fast sensory dynamics would result in more efficient processing when the perceptual system is confronted with taxing conditions such as fast stimulus presentation rates or background noise. We speculate that the correlation between speech task performance and task-dependent vMGB modulation might be a result of feedback from cerebral cortex areas. Feedback may emanate directly from auditory primary or association cortices, or indirectly via other structures such as the reticular nucleus with its inhibitory connections to the MGB (*Rouiller and de Ribaupierre, 1985*). Feedback cortico-thalamic projections from layer six in A1 to the vMGB, but also from association cortices such as the motion sensitive part of the planum temporale (*Tschentscher et al., 2019*), may modulate information ascending through the lemniscal pathway, rather than convey information to the ventral division (*Lee, 2013*; *Llano and Sherman, 2008*).

Although most of speech and language research focuses on cerebral cortex structures, investigating subcortical sensory contributions to speech perception is paramount to the development of a mechanistic understanding of how the human brain accomplishes speech recognition. The present study brings us a decisive step further in this direction by suggesting that it is the *primary* sensory auditory thalamus that shows a behaviorally relevant task-dependent modulation for speech recognition.

## Materials and methods

### Participants

The Ethics committee of the Medical Faculty, University of Leipzig, Germany approved the study (protocol number 273/14-ff). We recruited 33 participants using the database of the Max Planck Institute for Human Cognitive and Brain Sciences, Leipzig, Germany. The participants were right handed (as assessed by the Edinburgh Handedness Inventory; *Oldfield, 1971*), native German-speakers, had a mean age and standard deviation (SD) of 24.9 ± 2.5 years, and included 23 females. Participants provided written informed consent. None of the participants reported a history of psychiatric or neurological disorders, hearing difficulties, or current use of psychoactive medications. Normal hearing abilities were confirmed with pure tone audiometry (250 Hz to 8000 Hz) with a threshold equal to and below 25 dB (Madsen Micromate 304, GN Otometrics, Denmark). To exclude possible undiagnosed dyslexics, we tested the participant's reading speed and reading comprehension using the German LGVT: 6–12 test (*Schneider et al., 2007*). The cut-off for both reading scores were set to those levels mentioned in the test instructions as the 'lower average and above' performance range (i.e., 26–100% of the calculated population distribution). None of the participants performed below the cut-off performance (mean and standard deviation: 69.9 ± 19.5%, lowest mean score: 36%). Furthermore, none of the participants exhibited a clinically relevant number of traits associated with autism spectrum disorder as assessed by the Autism Spectrum Quotient (AQ; mean and standard deviation: 16.2 ± 4.8; cutoff: 32–50; *Baron-Cohen et al., 2001*). We tested AQ as autism can be associated with difficulties in speech-in-noise perception (*Alcántara et al., 2004*; *Groen et al., 2009*) and has overlapping symptoms with dyslexia (*White et al., 2006*). Participants received monetary compensation for participating in the study.

## Experiments

We performed three different functional MRI measurements: the speech experiment (n = 33), a MGB localizer (n = 33), and a tonotopy localizer (n = 28, 18 females, age 24.8 ± 5.0 years). Each experiment was performed once.

## Stimuli

### MGB and Tonotopy localizer

The stimuli for the MGB localizer and the tonotopy localizer consisted of 84 and 56 natural sounds, respectively, sampled at 16 kHz at 32 bit, and included samples of human speech, animal cries and tool sounds (these were the same as described in *Moerel et al., 2015*). The stimuli had a duration of 1000 ms, were ramped with 10 ms linear slopes, and had equalized root-mean-square levels.

### Speech experiment

The speech experiment stimuli consisted of 448 vowel-consonant-vowel (VCV) syllables with an average duration and SD of 803 ± 105 ms. These were spoken by three female and three male speakers (mean age and SD 27.7 ± 3.3 years) unfamiliar to the participants, and were recorded with a video camera (Canon Legria HFS10, Canon, Japan) and a Røde NTG-1 Microphone (Røde Microphones, Silverwater, NSW, Australia) connected to a pre-amplifier (TubeMP Project Series, Applied Research and Technology, Rochester, NY, USA) in a sound-attenuated room. The sampling rate was 48 kHz at 16 bit. Auditory stimuli were cut and flanked by Hamming windows of 15 ms at the beginning and end, converted to mono, and root-mean-square equalized using Python 3.6 (Python Software Foundation, www.python.org).

## Procedure

### MGB and Tonotopy localizer

For the MGB localizer and the tonotopy localizer, participants listened to natural sounds (human voices, animal cries, tool sounds; *Moerel et al., 2015*). The MGB localizer consisted of one run where 84 natural sound stimuli were presented in random order and had a duration of 12:50 min. The tonotopy localizer consisted of six runs where 56 of the 84 natural sound stimuli from the MGB localizer were presented. The sounds were randomly chosen before the first run and the same 56 sounds were played in each run. Each run had a duration of 8:58 min. To ensure listener engagement, in both localizers the participants performed a 1-back task and pushed a button when two consecutive sounds were the same. This happened on average 5% of the time. Additionally, 5% of the trials contained no sound (null events). Within each run, sounds were randomly jittered at an interval of 2, 3, or 4 repetition times (TR) and presented in the middle of the silent gap of 1200 ms (*Figure 1A*). The MGB localizer was used as an independent functional identifier for the left and right MGB. The resulting masks were then used to constrain the analyses of the tonotopy localizer to these regions of interest. In turn, the tonotopic regions of the MGB were used as masks in the speech experiment (see section Functional MRI Data Analysis).

### Speech experiment

In the speech experiment (*Figure 1C*) participants listened to blocks of auditory VCV syllables, and were asked to perform two types of tasks: a speech task and a speaker task. In the speech task, participants reported via button press whether the current syllable was different from the previous one (1-back task). In the speaker task, participants reported via button press whether the current speaker was different from the previous one. Speakers within a block were either all male or all female. This was necessary to avoid that participants performed a gender discrimination task on some trials and a speaker identity task on other trials. Task instructions were presented for two seconds prior to each block and consisted of white written words on a black background (German words 'Silbe' for syllable, and 'Person' for person). After the instruction, the block of syllables started (*Figure 1B*). Each block contained 14 stimuli. Each stimulus presentation was followed by 400 ms of silence. Within one block both syllables and speakers changed six or seven times. The average length of a block and SD was 17.0 ± 0.9 s. Counterbalancing of the stimulus material for the two tasks was achieved by presenting each block twice: once with the instruction to perform the speech task and once with the instruction to perform the speaker task. Besides the factor 'task', the experiment

included another factor. That is, blocks had either only vowel or only consonant changes. While this factor is included in the analysis, it is irrelevant for addressing the current research question.

The experiment was divided into five runs with a duration of 8:30 min per run. Each of the four condition blocks (speech vowel change, speaker vowel change, speech consonant change, speaker consonant change) were presented six times in pseudo-randomized order. The last stimulus presentation in the run was followed by 30 s of no stimulation. Participants were allowed to rest for one minute between runs. To familiarize participants with speakers' voices and to ensure they understood the task, they performed two initial training runs outside the MRI-scanner: one run for speaker familiarization, and one for experiment familiarization (detailed below).

The experiments were programmed and presented using Presentation (v17.1, NeuroBehavioral Systems, Berkley, CA, USA) in Windows XP and delivered through an MrConfon amplifier and earbuds linked to the transducers via air tubes (manufactured 2008, MrConfon GmbH, Magdeburg, Germany).

## Participant training

The participant training consisted of a speaker familiarization and an experiment familiarization.

The speaker familiarization consisted of a presentation of the speakers and a test phase. In the presentation phase, the speakers were presented in six blocks, each containing nine pseudo-randomly chosen stimuli. Participants heard a pseudo-random choice of the same stimuli used in the experiment. Each block contained one speaker-identity only. Participants were alerted to the onset of a new speaker-identity block with the phrase 'Andere/r Sprecher/in' (German for 'Another Speaker'). Participants listened to the voices with the instruction to memorize the speaker's voice. In the following test phase participants were presented with four blocks of nine trials that each contained syllable pairs spoken by the same or different speaker from the ones they were trained on. Participants were asked to indicate whether the speakers of the two syllables were the same by pressing keypad buttons '1' for yes and '2' for no. Participants received visual feedback for correct (green flashing German word for correct: 'Richtig') and incorrect (red flashing German word for incorrect: 'Falsch') answers. If participants scored below 85%, they repeated the speaker familiarization training.

The experiment familiarization consisted of one 8:30 min long run of the fMRI speech experiment (*Figure 1B/C*). Stimuli were randomly chosen from the same stimulus material used in the experiment. If participants scored below 85% across tasks in the experiment familiarization run they repeated the experiment familiarization training.

The training (speaker and experiment familiarization) took place within the same testing-session as the pre-tests (audiometry, reading comprehension, and AQ questionnaire), and was repeated half an hour prior to the fMRI experiment, which took place at a later date.

## Data acquisition and processing

MRI data were acquired using a Siemens Magnetom 7 T scanner (Siemens Healthineers, Erlangen, Germany) with a Nova 32-channel head coil (Nova Medical, Wilmington MA, USA). Functional MRI data were acquired using echo planar imaging (EPI) sequences. We used a field of view (FoV) of 132 × 132 mm and partial coverage with 28 slices. This volume was oriented obliquely such that the slices encompassed the inferior colliculi (IC), the MGB and the superior temporal gyrus, running in parallel to the latter (*Figure 6—figure supplement 1*).

The MGB and tonotopy localizers had the following acquisition parameters: TR = 2800 ms (acquisition time TA = 1600 ms, silent gap: 1200 ms), TE = 22 ms, flip angle 65°, GRAPPA (*Griswold et al., 2002*) with acceleration factor 2, 33% phase oversampling, matrix size 120 × 120, FoV 132 mm x 132 mm, phase partial Fourier 6/8, voxel size 1.1 mm x 1.1 mm x 1.1 mm, interleaved acquisition, anterior to posterior phase-encode direction. We employed a clustered EPI technique allowing for stimulus presentations in quiet in a fast-event related design. Stimuli were presented during the silent gap. For the MGB localizer, we acquired one run of 275 volumes (13:06 min). For the tonotopy, localizer we acquired 187 volumes (9:01 min) per run with a total of six runs.

For the speech experiment, acquisition parameters were the same as for the localizers, with the exception of a shorter TR (1600 ms) due to continuous scanning (i.e., no silent gap), 320 volumes per run, and total length of acquisition per run of 8:30 min. Five runs were recorded for each participant.

The acquisition parameters were similar to the protocol described by *Moerel et al. (2015)*, with the exception of a longer echo time and phase oversampling which eschewed front-back wrapping artifacts. An example EPI is shown in *Figure 6—figure supplement 2*. The difference in echo time between our sequence and the one in *Moerel et al. (2015)* may have resulted in a lower signal-to-noise ratio in subcortical structures. However, as the MGB has a T2* value of ~33 ms the different echo times of 19 ms (our sequence) and 22 ms (*Moerel et al., 2015*) had little to no effect on the applied general linear model (*de Hollander et al., 2017*).

During functional MRI data acquisition we also acquired physiological values (heart rate, and respiration rate) using a BIOPAC MP150 system (BIOPAC Systems Inc, Goleta, CA, USA). Structural images were recorded using an MP2RAGE (*Marques et al., 2010*) T1 protocol: 700 µm isotropic resolution, TE = 2.45 ms, TR = 5000 ms, TI1 = 900 ms, TI2 = 2750 ms, flip angle 1 = 5°, flip angle 2 = 3°, FoV 224 mm ×224 mm, GRAPPA acceleration factor 2, duration 10:57 min.

## Behavioral data analysis

Button presses were modeled using a binomial logistic regression which predicts the probability of correct button presses based on four independent variables (speech task, vowel change; speech task, consonant change; speaker task, vowel change; speaker, task consonant change) in a Bayesian framework (*Mcelreath, 2015*).

To pool over participants and runs we modeled the correlation between intercepts and slopes. For the model implementation and data analysis, we used PyMC3 (*Salvatier et al., 2016*) using a No-U-Turn Sampler (*Hoffman and Gelman, 2011*) with three parallel chains. Per chain we had 20,000 samples with 5000 of these as warm-up. Only the latter 7500 were used for posterior mean and highest posterior density (HPD) interval estimates. The difference in percent correct button presses between the speech and speaker task was calculated using the posterior densities averaged over consonant and vowel changes. The resulting distribution was averaged and the 97% HPD was calculated. If the posterior probability distribution does not strongly overlap zero, then there was a detectable difference between conditions (*Mcelreath, 2015*).

The predictors included in the behavioral data model were: task (1 = speech, 0 = speaker), and syllable change (1 = vowel, 0 = consonant). We also included the two way interaction of task and syllable change. Because data were collected across participants and runs it is reasonable to include random effects for both of these in the logistic model, albeit we expected little to no difference in performance between runs. The model had the following likelihood and linear models:

$$L_i \sim Binomial(1, p_{i,j})$$

$$\text{logit}(p_{i,j}) = A_{i,j} + B_{S,i,j} + B_{V,i,j} + B_{SV,i,j}, \text{for } i = 1, \ldots, I; j = 1, \ldots, J$$

$$A_{i,j} = \alpha + \alpha_{\text{participant}[i]} + \alpha_{\text{run}[j]}$$

$$B_{S,i,j} = \beta_S + \beta_{S, \text{participant}[i]} + \beta_{S, \text{run}[j]}$$

$$B_{V,i,j} = \beta_V + \beta_{V, participant[i]} + \beta_{V, \text{run}[j]}$$

$$B_{SV,i,j} = \beta_{SV} + \beta_{SV, \text{participant}[i]} + \beta_{SV, \text{run}[j]}$$

$$\begin{bmatrix} \alpha_{participant} \\ \beta_{S,\text{participant}} \\ \beta_{V,\text{participant}} \\ \beta_{SV,\text{participant}} \end{bmatrix} \sim \text{MVNormal}\left( \begin{bmatrix} \alpha \\ \beta_S \\ \beta_V \\ \beta_{SV} \end{bmatrix}, \mathbf{S}_{participant} \right)$$

$$\begin{bmatrix} \alpha_{\text{run}} \\ \beta_{S,\text{run}} \\ \beta_{V,\text{run}} \\ \beta_{SV,\text{run}} \end{bmatrix} \sim \text{MVNormal}\left( \begin{bmatrix} \alpha \\ \beta_S \\ \beta_V \\ \beta_{SV} \end{bmatrix}, \mathbf{S}_{\text{run}} \right)$$

$$\mathbf{S_{participant}} = \begin{pmatrix} \sigma_\alpha & 0 & 0 & 0 \\ 0 & \sigma_{\beta_S} & 0 & 0 \\ 0 & 0 & \sigma_{\beta_S} & 0 \\ 0 & 0 & 0 & \sigma_{\beta_S V} \end{pmatrix} \mathbf{R_{participant}} \begin{pmatrix} \sigma_\alpha & 0 & 0 & 0 \\ 0 & \sigma_{\beta_S} & 0 & 0 \\ 0 & 0 & \sigma_{\beta_S} & 0 \\ 0 & 0 & 0 & \sigma_{\beta_S V} \end{pmatrix}$$

$$\mathbf{S_{run}} = \begin{pmatrix} \sigma_\alpha & 0 & 0 & 0 \\ 0 & \sigma_{\beta_S} & 0 & 0 \\ 0 & 0 & \sigma_{\beta_S} & 0 \\ 0 & 0 & 0 & \sigma_{\beta_S V} \end{pmatrix} \mathbf{R_{run}} \begin{pmatrix} \sigma_\alpha & 0 & 0 & 0 \\ 0 & \sigma_{\beta_S} & 0 & 0 \\ 0 & 0 & \sigma_{\beta_S} & 0 \\ 0 & 0 & 0 & \sigma_{\beta_S V} \end{pmatrix}$$

$$\alpha \sim \mathrm{Normal}(0,5)$$

$$\beta_S \sim \mathrm{Normal}(0,5)$$

$$\beta_V \sim \mathrm{Normal}(0,5)$$

$$\beta_{SV} \sim \mathrm{Normal}(0,5)$$

$$(\sigma_{\mathrm{participant}}, \sigma_{\mathrm{run}}) \sim \mathrm{HalfCauchy}(2)$$

$$\sigma_{corr,\,participant} \sim \mathrm{HalfCauchy}(2)$$

$$\sigma_{corr,\,run} \sim \mathrm{HalfCauchy}(2)$$

$$\mathbf{R_{participant}} \sim LKJcorr\left(4, \sigma_{corr,\,participant}\right)$$

$$\mathbf{R_{run}} \sim LKJcorr\left(4, \sigma_{corr,run}\right)$$

where $I$ is the number of subjects and $J$ is the number of runs. The model is compartmentalized into sub-models for the intercept and each slope. $A_{i,j}$ is the sub-model for the intercept for observations $i,j$. Similarly, $B_{S,i,j}$, $B_{V,i,j}$ and $B_{SV,i,j}$ are the sub-models for the speech-speaker slope, vowel-consonant slope and the interaction slope; $S_{participant}$ and $S_{run}$ are the covariance matrices, $R_{participant}$ and $R_{run}$ are the priors for the correlation matrices between the intercepts and slopes modeled as a LKJ probability density for participants and runs (*Lewandowski et al., 2009*). Informative priors for the intercept ($\alpha$) and additional coefficients (e.g., $\beta_S$), random effects for subject and run ($\beta_{S,\,participant}, \beta_{S,run}$), and multivariate priors for subjects and runs identify the model by constraining the position of $p_{i,j}$ to reasonable values.

To check for correlations in the performance between tasks, we calculated the Pearson's product moment (*Cohen, 1988* p. 75) on the proportion of hits in the speech and speaker task using Python 3.6.

We analyzed reaction times using a linear model in a Bayesian framework. Reaction times were mean centered and the priors on reaction times were modeled as T-distributions. We calculated the mean difference in reaction times between the speech and the speaker task to check for a speed-accuracy trade-off. The model is described below:

$$L_i \sim Binomial(1, p_{i,j})$$

$$p_{i,j} = A_{i,j} + B_{S,i,j} + B_{V,i,j} + B_{SV,i,j}, \text{for } i = 1, \dots, I; j = 1, \dots, J$$

$$A_{i,j} = \alpha + \alpha_{\mathrm{participant}[i]} + \alpha_{\mathrm{run}[j]}$$

$$B_{S,i,j} = \beta_S + \beta_{S,\,\mathrm{participant}[i]} + \beta_{S,\,\mathrm{run}[j]}$$

$$B_{V,i,j} = \beta_V + \beta_{V,\,participant[i]} + \beta_{V,\,\mathrm{run}[j]}$$

$$B_{SV,i,j} = \beta_{SV} + \beta_{SV,\,\mathrm{participant}[i]} + \beta_{SV,\,\mathrm{run}[j]}$$

$$\begin{bmatrix} \alpha_{participant} \\ \beta_{S,participant} \\ \beta_{V,participant} \\ \beta_{SV,participant} \end{bmatrix} \sim \mathrm{MVNormal}\left( \begin{bmatrix} \alpha \\ \beta_S \\ \beta_V \\ \beta_{SV} \end{bmatrix}, \mathbf{S}_{participant} \right)$$

$$\begin{bmatrix} \alpha_{\mathrm{run}} \\ \beta_{S,\mathrm{run}} \\ \beta_{V,\mathrm{run}} \\ \beta_{SV,\mathrm{run}} \end{bmatrix} \sim \mathrm{MVNormal}\left( \begin{bmatrix} \alpha \\ \beta_S \\ \beta_V \\ \beta_{SV} \end{bmatrix}, \mathbf{S}_{\mathrm{run}} \right)$$

$$\mathbf{S}_{\mathbf{participant}} = \begin{pmatrix} \sigma_\alpha & 0 & 0 & 0 \\ 0 & \sigma_{\beta_S} & 0 & 0 \\ 0 & 0 & \sigma_{\beta_S} & 0 \\ 0 & 0 & 0 & \sigma_{\beta_S V} \end{pmatrix} \mathbf{R}_{\mathbf{participant}} \begin{pmatrix} \sigma_\alpha & 0 & 0 & 0 \\ 0 & \sigma_{\beta_S} & 0 & 0 \\ 0 & 0 & \sigma_{\beta_S} & 0 \\ 0 & 0 & 0 & \sigma_{\beta_S V} \end{pmatrix}$$

$$\mathbf{S}_{\mathbf{run}} = \begin{pmatrix} \sigma_\alpha & 0 & 0 & 0 \\ 0 & \sigma_{\beta_S} & 0 & 0 \\ 0 & 0 & \sigma_{\beta_S} & 0 \\ 0 & 0 & 0 & \sigma_{\beta_S V} \end{pmatrix} \mathbf{R}_{\mathbf{run}} \begin{pmatrix} \sigma_\alpha & 0 & 0 & 0 \\ 0 & \sigma_{\beta_S} & 0 & 0 \\ 0 & 0 & \sigma_{\beta_S} & 0 \\ 0 & 0 & 0 & \sigma_{\beta_S V} \end{pmatrix}$$

$$\alpha \sim T(3,0,1)$$

$$\beta_S \sim T(3,0,1)$$

$$\beta_V \sim T(3,0,1)$$

$$\beta_{SV} \sim T(3,0,1)$$

$$\left( \sigma_{\mathrm{participant}}, \sigma_{\mathrm{run}} \right) \sim \mathrm{HalfCauchy}(2)$$

$$\sigma_{corr,\,participant} \sim \mathrm{HalfCauchy}(2)$$

$$\sigma_{corr,\,run} \sim \mathrm{HalfCauchy}(2)$$

$$\mathbf{R}_{\mathbf{participant}} \sim LKJcorr\left( 4, \sigma_{corr,\,participant} \right)$$

$$\mathbf{R}_{\mathbf{run}} \sim LKJcorr\left( 4, \sigma_{corr,run} \right)$$

where $I$ is the number of subjects and $J$ is the number of runs. The model is compartmentalized into sub-models for the intercept and each slope. $A_{i,j}$ is the sub-model for the intercept for observations $i,j$. Similarly, $B_{S,i,j}$, $B_{V,i,j}$ and $B_{SV,i,j}$ are the sub-models for the speech-speaker slope, vowel-consonant slope and the interaction slope; $S_{participant}$ and $S_{run}$ are the covariance matrices, $R_{participant}$ and $R_{run}$ are the priors for the correlation matrices between the intercepts and slopes modeled as a LKJ probability density for participants and runs (*Lewandowski et al., 2009*). Informative priors for the intercept ($\alpha$) and additional coefficients (e.g., $\beta_S$), random effects for subject and run ($\beta_{S,\,participant}$, $\beta_{S,run}$), and multivariate priors for subjects and runs identify the model by constraining the position of $p_{i,j}$ to reasonable values.

## Functional MRI data analysis

### Preprocessing of MRI data

The partial coverage by the 28 slices and the lack of a whole brain EPI measurement resulted in coregistration difficulties of functional and structural data. As a workaround, the origin (participant space coordinate [0, 0, 0]) of all EPI and MP2RAGE images were manually set to the anterior commissure using SPM 12. Furthermore, to deal with the noise surrounding the head in MP2RAGE images, these were first segmented using SPM's new segment function (SPM 12, version 12.6906, Wellcome Trust Centre for Human Neuroimaging, UCL, UK, http://www.fil.ion.ucl.ac.uk/spm) running on Matlab 8.6 (The Mathworks Inc, Natick, MA, USA). The resulting gray and white matter segmentations were summed and binarized to remove voxels that contain air, scalp, skull and cerebrospinal fluid from structural images using the ImCalc function of SPM.

A template of all participants was created with ANTs (*Avants et al., 2009*) using the participants' MP2RAGE images, which was then registered to the MNI space using the same software package and the MNI152 (0.5 mm)$^3$ voxel size template provided by FSL 5.0.8 (*Smith et al., 2004*). All MP2RAGE images were preprocessed with Freesurfer (*Fischl et al., 2002*; *Fischl et al., 2004*; *Han and Fischl, 2007*) using the recon-all command to obtain boundaries between gray and white matter, which were later used in the functional to structural registration step.

The rest of the analysis was coded in nipype (*Gorgolewski et al., 2011*). A graphical overview of the nipype pipeline can be found in *Figure 6—figure supplement 3*. Head motion and susceptibility distortion by movement interaction of functional runs were corrected using the Realign and Unwarp method (*Andersson et al., 2001*) in SPM 12 after which outlier volumes were detected using ArtifactDetect (composite threshold of translation and rotation: 1; intensity Z-threshold: 3; global threshold: 8; https://www.nitrc.org/projects/artifact_detect/). Coregistration matrices for realigned functional runs per participant were computed based on each participant's structural image using Freesurfer's BBregister function (register mean EPI image to T1, option '—init-header' was specified in order to preserve the origin of the manual alignment of structural and functional data). Warping using coregistration matrices (after conversion to ITK coordinate system) and resampling to 1 mm isovoxel was performed using ANTs. Before model creation we smoothed the data in SPM12 using a 1 mm kernel at full-width half-maximum.

## Physiological data

Physiological data (heart rate and respiration rate) were processed by the PhysIO Toolbox (*Kasper et al., 2017*) to obtain Fourier expansions of each, in order to enter these into the design matrix (see statistical analyses sections below).

## Statistical analysis of the speech experiment

Models were set up in SPM using the native space data for each participant. The design matrix included three cardiac and four respiratory regressors, six realignment parameters, and a variable number of outlier regressors from the ArtifactDetect step, depending on how many outliers were found in each run. These regressors of no interest were also used in the models of the other two experiments (MGB and tonotopy localizer). Since participants provided a response only for the target stimulus changes and not for each stimulus presentation, we modeled these to eschew a potential sensory-motor confound as 0.5 for hit, –0.5 for miss and 0.0 for everything else. If more than one syllable presentation took place within one volume acquisition, the values within this volume were averaged. The speech experiment had a total of five modeled conditions, which were convolved with the hemodynamic response function (HRF): speech task/vowel change, speech task/consonant change, speaker task/vowel change, speaker task/consonant change, and task instruction. Parameter estimates were computed for the contrast speech vs speaker at the first level using restricted maximum likelihood (REML) as implemented in SPM 12.

After estimation, the contrasts were registered to the MNI structural template of all participants using a two-step registration in ANTs (see also *Figure 6—figure supplement 3*). First, a quick registration was performed on the whole head using rigid, affine and diffeomorphic (using Symmetric Normalization: SyN) transformations and the mutual information similarity metric. Second, the high quality registration was confined to a rectangular prism mask encompassing the left and right MGB, and IC only. This step used affine and SyN transformations and mean squares and neighborhood

cross correlation similarity measures, respectively. We performed the registration to MNI space for all experiments by linearly interpolating the contrast images using the composite transforms from the high quality registration.

We used a random effects (RFX) analysis to compute the speech vs speaker contrast across participants to test our first hypothesis that the MGB response is modulated by this contrast. To do this, we took the first level contrasts across participants and entered them into an RFX model to be estimated using REML. Based on the results of previous experiments (*Díaz et al., 2012*; *von Kriegstein et al., 2008b*), we expected a result for the categorical speech vs speaker contrast in the left and right MGB. Our second hypothesis was that the proportion of correct button presses in the speech task correlates with the responses elicited by the speech vs speaker contrast over participants in the left MGB only (*von Kriegstein et al., 2008b*; *Díaz et al., 2012*). We thus computed the RFX correlation between the speech vs speaker contrast and the proportion of correct button presses in the speech task across participants. This was implemented using the behavioral percent correct scores for the speech task as a covariate of interest for each participant in the SPM RFX model. We used an equivalent procedure to test for correlation between the speech vs speaker task contrast and the proportion of correct button presses in the speaker task across participants. To formally test for a difference between these two correlations we performed a comparison using a freely available online tool (comparingcorrelations.org) (*Diedenhofen and Musch, 2015*). This tool computes the z-score and p-value of the difference between two correlations based on *Hittner et al. (2003)*.

## Meta-analysis of the main effect of task (speech vs speaker task contrast)

The lack of statistical significance for the speech vs speaker contrast raised the question whether the overall effect is different from the ones reported previously (*Díaz et al., 2012*; *von Kriegstein et al., 2008b*). We performed a random effects meta-analysis to test whether the lack of task-dependent modulation in the present study was different from other studies that have reported a task-dependent modulation of the MGB. We included five studies in the meta-analysis that included a speech task vs control task contrast: two experiments from *von Kriegstein et al. (2008b)*, the data from the control participants of *Díaz et al. (2012)*, the result of a recent study (*Mihai et al., 2019*), and the result of the current study. Effect sizes and standard errors were entered into a random effects model that was estimated with maximum likelihood using JASP 0.9 (jasp-stats.org).

## Statistical analysis of the MGB localizer

For the MGB localizer we used a stick function convolved with the HRF to model each presented sound. Null events were not modeled, as well as repeated sounds, to avoid a sensory-motor confound through the button-press. The data were modeled according to *Perrachione and Ghosh (2013)* where repetition (TR = 2.8 s) and acquisition times (TA = 1.6 s) were modeled separately. The contrast *Sound vs Silence* was computed for each participant. The inference across participants was modeled using the first level contrasts in a second-level RFX analysis for the group. Significant voxels (see Section Masks below) in the left and right MGB found in the RFX analysis for the contrast sound vs silence were used as a mask for the tonotopy localizer.

## Statistical analysis of the tonotopy localizer

For the tonotopy localizer we followed a similar approach as *Moerel et al. (2015)*. The sounds were first processed through the NSL toolbox (*Chi et al., 2005*) which mimics the spectral transformation of sounds passing through the cochlea to the midbrain. This frequency representation includes a bank of 128 overlapping bandpass filters equally spaced on a log frequency axis (180-7040 Hz; range 5.3 octaves). The resulting spectrograms were averaged over time. To reduce overfitting we divided the tonotopic axis into 12 equal bandwidths in octaves and averaged the model's output within these regions. The MrConfon headphones guarantee a linear frequency response up to 4 kHz, thus only the first 10 bins were used in the analysis, which resulted in 10 frequency bins for each sound file. The frequency model consisted of a vector of values corresponding to the frequency representations per sound. Since each sound had a frequency representation the final model is a matrix $W = [S \times F]$, where $S$ is the number of sounds and $F$ the number of features per sound. The predictors were z-scored across bins since low frequencies have more energy and would thus be

more strongly represented compared to high frequencies (*Moerel et al., 2015*). The matrix was convolved with the hemodynamic response function and its components (i.e., the 10 frequency bins) were used as regressors of interest in the design matrix of SPM. In addition, we included the same regressors of no-interest as in the design matrix for the speech experiment (i.e., six respiratory regressors, six realignment parameters, and a variable number of outlier regressors from the ArtifactDetect step, depending on how many outliers were found). Parameter estimates were calculated for each frequency bin at the first level in native space.

## Masks

### MGB localizer

We created masks using all voxels from the second level MGB localizer analysis for the contrast sound vs silence (family-wise error [FWE] corrected $p<0.001$) constrained within a $r = 5$ mm sphere centered at the voxel with the statistical maximum in the left and right MGB. We chose such a stringent p-value due to the strong effect and the multitude of above threshold voxels found within and around the left and right MGB. This procedure excluded all voxels which were clearly too far away from the structural boundaries of the MGB as seen in the MP2RAGE MNI template, yet still within the cluster, to be considered part of the MGB. These masks were inverse transformed per participant from MNI space to participant space using ANTs. Above threshold voxels (uncorrected $p<0.05$) within the transformed masks were extracted, for each participant, from the MGB localizer *Sound vs Silence* contrast. These masks were then used to define each participant's tonotopy with the tonotopy localizer.

### Tonotopy localizer

Each voxel within each participant's left and right MGB and IC localizer mask was labeled according to the frequency bin to which it responded strongest, that is which had the highest parameter estimate (*Moerel et al., 2015*). Thus, voxels would have values from 1 to 10 corresponding to the frequency bin that they best represented. This resulted in a map of frequency distributions from low to high frequencies in the left and right MGB for each participant. To create masks at the group level, these tonotopic maps were registered to MNI space using ANTs and averaged across participants.

To evaluate the tonotopic representations in the MGB and IC in a similar way as *Moerel et al. (2015)*, we visually inspected the direction which showed the strongest tonotopy. This was a dorsal-lateral to ventral-medial gradient that was most visible in a sagittal view. We thus rotated and resliced the individual maps around the z-axis by $90°$, which placed the sagittal view in the x-y plane. In this plane we calculated gradient directions in 10 adjacent slices, ensuring a representative coverage of the tonotopic pattern. A cut at $90°$ captured both low and high frequency areas. Histograms in 5° steps were calculated for each slice. The histograms of the gradients were then averaged first over slices per participant, followed by an average over participants. Based on the atlas by *Morel et al. (1997)* and findings of MGB subdivisions in awake primates (*Bartlett and Wang, 2011*) we parcellated the resulting frequency gradients as distinct regions. Voxels that represented the highest frequency were chosen as the boundary within each slice. Voxels above this boundary corresponded to one region, and those below this boundary to the other region. The regions were drawn in each slice using ITKSnap (v. 3.6.0; *Yushkevich et al., 2006*). Volume size and center of mass (COM) for each gradient are listed in *Table 1*.

For the IC, we created functional masks of IC responses from the MGB localizer experiment (Sound vs Silence, uncorrected $p<0.05$) and constrained these to the anatomical volumes of the IC. Frequency distribution maps were calculated per participant in MNI space and averaged. Gradient directions were calculated from the mean tonotopy maps in three different slices. Histograms were then averaged and plotted. As the IC was not part of the main objective of this manuscript, we report it in the *Supplementary file 1* and *Figure 3—figure supplement 1*.

Unthresholded t-maps of contrasts of interest, tonotopy maps of the MGB and IC, as well as vMGB masks are available on neurovault (https://neurovault.org/collections/4785/).

## Significance testing

We used small volume corrections (SVC) to test for significant voxels for the contrast speech vs speaker task as well as the correlation of speech vs speaker task with the behavioral proportion

correct scores in the speech task (significance defined as $p<0.05$ FWE corrected for the region of interest). We tested bilaterally using the vMGB masks described above for the first hypothesis (main effect of task) and left vMGB for the second hypothesis (correlation between speech recognition performance and main effect of task) motivated by findings in previous studies (*von Kriegstein et al., 2008a*; *Díaz et al., 2012*).

# Additional information

### Funding

| Funder | Grant reference number | Author |
|---|---|---|
| Max-Planck-Gesellschaft | Research Group Grant and open-access funding | Paul Glad Mihai Katharina von Kriegstein |
| H2020 European Research Council | SENSOCOM (647051) | Paul Glad Mihai Katharina von Kriegstein |
| Max-Planck-Gesellschaft | | Robert Trampel |
| Nederlandse Organisatie voor Wetenschappelijk Onderzoek | VIDI grant 864-13-012 | Federico de Martino |
| Nederlandse Organisatie voor Wetenschappelijk Onderzoek | VENI grant 451-15-012 | Michelle Moerel |

The funders had no role in study design, data collection and interpretation, or the decision to submit the work for publication.

### Author contributions

Paul Glad Mihai, Conceptualization, Resources, Data curation, Software, Formal analysis, Investigation, Visualization, Methodology, Writing—original draft, Project administration, Writing—review and editing; Michelle Moerel, Federico de Martino, Robert Trampel, Stefan Kiebel, Methodology, Writing—review and editing; Katharina von Kriegstein, Conceptualization, Supervision, Funding acquisition, Methodology, Writing—original draft, Project administration, Writing—review and editing

### Author ORCIDs

Paul Glad Mihai [ID] https://orcid.org/0000-0001-5715-6442
Katharina von Kriegstein [ID] https://orcid.org/0000-0001-7989-5860

### Ethics

Human subjects: The Ethics committee of the Medical Faculty, University of Leipzig, Germany, approved the study (protocol number 273/14-ff). Participants provided written informed consent to participate in the study and publish the material in a scientific journal.

### Decision letter and Author response

Decision letter https://doi.org/10.7554/eLife.44837.024
Author response https://doi.org/10.7554/eLife.44837.025

# Additional files

### Supplementary files

• Supplementary file 1. Montreal neurological institute (MNI) coordinates, p-values, T-values and parameter estimates (β) and 90% confidence intervals (CI) for voxels within regions for which we did not have an a-priori hypothesis. Family-wise error corrected p-values were calculated using small volume correction for the voxels within the masks.
DOI: https://doi.org/10.7554/eLife.44837.020

• Transparent reporting form

DOI: https://doi.org/10.7554/eLife.44837.021

## Data availability

As participants did not give consent for their functional MRI data to be released publicly within the General Data Protection Regulation 2016/679 of the EU, these data can be made available on request to the corresponding author. Behavioral data are found in Figure 5-source data 1. Unthresholded maps have been uploaded to neurovault.org for the contrasts speech vs speaker task and the correlation speech vs speaker task with proportion of hits in the speech task (Figure 6). Additionally, the tonotopic maps for left and right MGB and IC are also provided (Figures 2, 3, and Figure 3-figure supplement 1), as well as the left and right vMGB mask (Figure 6). Files can be found at: https://neurovault.org/collections/4785/.

The following dataset was generated:

| Author(s) | Year | Dataset title | Dataset URL | Database and Identifier |
|---|---|---|---|---|
| Paul Glad Mihai, Michelle Moerel, Federico de Martino, Robert Trampel, Stefan Kiebel, Katharina von Kriegstein | 2019 | Modulation of tonotopic ventral MGB is behaviorally relevant for speech recognition | https://neurovault.org/collections/4785/ | NeuroVault, 4785 |

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
