## [Decision Letter]

Thank you for submitting your article "Modulation of tonotopic ventral MGB is behaviorally relevant for speech recognition" for consideration by *eLife*. Your article has been reviewed by three peer reviewers, including Timothy D Griffiths as the Reviewing Editor and Reviewer #1, and the evaluation has been overseen by Andrew King as the Senior Editor. The following individual involved in review of your submission has agreed to reveal their identity: Matthew H Davis (Reviewer #3).

The reviewers have discussed the reviews with one another and the Reviewing Editor has drafted this decision to help you prepare a revised submission.

The work is well executed in seeking changes for a controlled contrast between a speech and non-speech task in vMGB, could fit with a predictive coding account, which would be interesting and novel in applying predictive coding accounts to subcortical structures. The main issue is that more robust support for the conclusion that the MGB is involved in processing speech is required: the reviewers were concerned about the absence of a straightforward main effect for the speech minus non-speech task, the weak association between accuracy scores in the speech task and MGB modulation, and the interpretation of the latter which was clearly not based on an a prior hypothesis. We felt further analyses to achieve this were possible within two months hence the decision to ask for a major revision.

The individual reviews are appended below, but the major changes required are summarised here by the Reviewing Editor:

1) Further analyses to combine the present null finding for the main effect of task with previous results as suggested by reviewer 3.

2) Further analysis to make the correlation between performance and activation in MGB more robust is also required.

*Reviewer #1:*

I think the work is well executed and very interesting in seeking changes for a nicely controlled contrast between a speech and non-speech task in vMGB, which I agree could fit with predictive coding account, and which would be interesting in applying predictive coding accounts to subcortical structures which I think is novel in auditory work.

I had some questions for the authors:

1) I did not actually think the tonotopy data were required but I guess they do make the case they are examining a lemniscal area. I would be interested in whether the authors feel the results are consistent with those in macaque (PMID 21378972). The gradient running anteromedially looks similar to me and I would not expect much difference (in contrast to the speech).

2) The existence of a significant speech minus non-speech contrast might have fitted a predictive coding account and was clearly predicted by the authors. I found the left vMGB correlation between the [speech vs speaker] contrast and speech recognition performance interesting but rather more nuanced in interpretation and I think that slightly reduces the impact of the work.

*Reviewer #2:*

The study examines the extent to which task-related modulation (speech>speaker focus using the same set of stimuli) is present within the ventral medial geniculate body (MGB), the section that from animal models and previous work in humans is known to be tonotopically organized. To achieve this goal, the authors run several different conditions at ultra-high field (7T): a tonotopy localizer that replicates a prior study Moerel et al., 2015, in a significant sample size (28 participants!) establishing tonotopic gradients within the MGB, and a behavioral task involving participants making judgments (one back) regarding change in speech content (consonant/vowel) or speaker. The results show clear tonotopic gradients-based on animal models and anatomical considerations of vMGB were identified. This is a direct replication of Moerel et al., 2015. The second part of the experiment involved examining task-related modulation of the identified vMGB, with the hypothesis that speech change>speaker change will evoke the left MGB (consistent with previous experiments from the group: von Kriegstein et al., 2008, Diaz et al., 2012). This prediction was invalidated by the results that show no task-related modulation (speech>speaker) in the vMGB. A potential confound here is that the speaker change condition was statistically harder (speaker judgments were within sex) than the speech change. This behavioral result differed from prior experiments from the same group. An additional correlation analysis was performed indicating a weak association between accuracy scores in the speech task and MGB modulation (greater task modulation relates to better speech performance). There is little variability however in the speech task performance (is the range from 84% to 95% biologically-relevant?).

Major comments: There are less than a handful of studies examining the human auditory thalamus at ultra-high fields-a severe paucity that limits our understanding of thalamic function in processing behaviorally-relevant signals in humans. Within this context, the goals of the present study are laudable. The study convincingly shows tonotopic representation in the vMGB, providing a large-scale replication of Moerel et al., 2015. I'm not at all convinced about the other main thrust of the study related to task modulation/speech recognition. The results show no statistically significant difference in vMGB activity as a function of task. The correlation analyses yield a weak association, but the nature of this association is unclear. What does it mean that modulation in the left vMGB correlates with the speech recognition scores when the task performance in speech vary very little, and there are substantial accuracy differences between the speech and the speaker task? The authors provide three potential explanations for their lack of replication of prior work and seem to suggest that the most likely explanation is that MGB modulation might also have played a role in performing the speaker task. I agree, but this also suggests a critical flaw with the study design which a priori defined task modulation on the basis of a difference in the speech>speaker contrast.

*Reviewer #3:*

This paper describes a high resolution fMRI study to establish whether a tonotopic, auditory region of the thalamus (ventral MGB) corresponds to the thalamus region previously shown to have activity modulated during performance of a speech perception task (compared to a well matched speaker identification task), and to correlate with individual differences in speech perception performance. The study appears well constructed and uses an impressive set of state-of-the-art methods including ultra-high field 7T fMRI, sub-mm voxels and sparse imaging, tonotopic modelling to specify frequency preferences and between voxel analysis to identify tonotopic gradients, etc.

However, despite these many technical strengths, I didn't feel that the present manuscript led to a substantial advance in my understanding of the contributions made by the auditory thalamus to speech perception. In part this is due to a surprising failure to replicate several previous findings of additional activation for a speech identification than a speaker identification task (from three similar conventional 3T fMRI paradigms reported previously by the same group). Both for this result, and the cross-subject correlation of activation with performance that is replicated in the present study, I felt that additional statistical analysis would allow the authors to strengthen their interpretation of their findings. At present, too many of the conclusions of the study appear to be speculation in the absence of statistical evidence. I'll expand on this below:

1) Speech > speaker task activation in vMGB – the authors remark that their failure to observe this effect is: "surprising, as this categorical task-effect was observed in three previous experiments in participants with typical development (von Kriegstein et al., 2008, Díaz et al., 2012)." Then in the Discussion section they offer three potential factors that might mediate this surprising outcome.

However, all of this discussion is building on a null finding. The absence of evidence for a speech task effect is not evidence of absence. Further analysis is necessary to show which of two interpretations is more likely: (1) that this is an underpowered or merely unlucky study, or (2) a true change from the expected positive to a null (or negative?) effect. Positive evidence of a change in the magnitude of the effect could come from a between-experiment comparison (using activation vs rest as a dependent measure to ensure cross-experiment findings are comparable despite differences in field strength, experimental design, etc.). Or they could use a Bayesian or meta-analytic method to determine whether the expected, opposite or null hypothesis is more likely for the current experiment when combined with a prior from previous studies. Without these analyses, though, the three paragraphs of speculation regarding possible causes of a change in experimental outcome are irrelevant. No compelling statistical evidence for a change has been provided.

2) Correlation with individual differences in speech perception performance – despite the lack of a categorical task effect, the authors use their correlation with behaviour as evidence that: "confirmed our hypothesis that the left first-order thalamic nucleus – vMGB – is involved in speech recognition." Later in the same paragraph they go on to argue that: "The results can be explained neither by differences in stimulus input in the two conditions, as the same stimuli were heard in both tasks, nor by a correlation with general better task performance, as there was no correlation with the speaker task."

Here again, they are arguing for a difference between a positive finding (correlation) and a null finding (no correlation) based on the difference between a p-value less than 0.05 and another above 0.05. This is fallacious. They should statistically compare the two correlations to confirm that there is a difference between the speech and speaker findings in order to confirm that some task-specific, auditory aspect of speech perception, rather than generic attentional, executive or motoric factors (that can also be associated with performance) are linked to these regions of the auditory thalamus.

Summary: These two stringent criticisms challenge the authors to do more with the data that they have collected. Nonetheless, I think that the work that these authors have done to localise the tonotopic regions of the auditory thalamus has considerable merit (though it is too far outside my main interest for me to assess this contribution in detail). Yet, I still think further statistical evidence is required for them to associate this thalamus region with a single, specifically-auditory aspect of speech perception (and not speaker perception or generic task abilities). I'd like to see the authors do this additional work in order to more convincingly establish the ventral MGB as part of the neural circuitry for speech perception.

---

## [Author Response]

The work is well executed in seeking changes for a controlled contrast between a speech and non-speech task in vMGB, could fit with a predictive coding account, which would be interesting and novel in applying predictive coding accounts to subcortical structures. The main issue is that more robust support for the conclusion that the MGB is involved in processing speech is required: the reviewers were concerned about the absence of a straightforward main effect for the speech minus non-speech task, the weak association between accuracy scores in the speech task and MGB modulation, and the interpretation of the latter which was clearly not based on an a prior hypothesis. We felt further analyses to achieve this were possible within 2 months hence the decision to ask for a major revision.

We thank the reviewers for the constructive and very helpful comments. Here, we want to emphasize that both the main effect of task as well as the correlation of the task-dependent modulation with speech recognition behavior were equally weighted a priori hypotheses based on prior findings (von Kriegstein et al., 2008; Diaz et al., 2012) (see detailed reply below). In addition, we were surprised that the association between accuracy scores in the speech tasks and MGB modulation was considered weak. According to Cohen (1988; p.88) the effect would be considered large. We have detailed this our response to reviewer #2’s comments.

The individual reviews are appended below, but the major changes required are summarised here by the Reviewing Editor:1) Further analyses to combine the present null finding for the main effect of task with previous results as suggested by reviewer 3.

We thank the reviewers for this great suggestion. We performed meta-analyses for the main effect that included results from three previous experiments (Díaz et al., 2012; von Kriegstein et al., 2008) and results from a recent study (Mihai et al., 2019). The results showed a positive large meta-analytic effect over five studies for the main effect of speech vs non-speech task. Four studies with overlapping confidence intervals showed a positive main effect of task and only the present study falls out of this scheme. Given this result and the large sample size of the present study, it is highly likely that there is a real difference in the main effect of task between the present and the other four studies. We have integrated the new analyses in the manuscript.

2) Further analysis to make the correlation between performance and activation in MGB more robust is also required.

We thank the reviewers for this constructive comment. We performed two additional analyses. In the first analysis, we excluded one data point that exceeded two standard deviations from the mean and recalculated the correlation of the speech vs speaker contrast with the behavioral performance in the speech task across participants. The correlation resulted in very similar values (T=2.92, p=0.038, r=0.47) as the correlation with the outlier included. Thus the correlation is robust to outlier removal and we have now also included it in the manuscript. Second, we revisited the control correlation and found that there is a difference in correlations (correlation 1: [speech vs speaker] correlated with proportion of hits in speech task; correlation 2: [speaker vs speech] correlated with proportion of hits in the speaker task). We however, also found a positive correlation ([speech vs speaker] correlated with proportion of hits in the speaker task). This result can be explained by a correlation in task-performance across the speech and speaker task. Thus, we cannot claim that the vMGB task-dependent modulation (speech vs speaker) is specific to the speech task performance and we have removed instances in the manuscript that might imply such a specificity claim.

Reviewer #1:I think the work is well executed and very interesting in seeking changes for a nicely controlled contrast between a speech and non-speech task in vMGB, which I agree could fit with predictive coding account, and which would be interesting in applying predictive coding accounts to subcortical structures which I think is novel in auditory work.I had some questions for the authors1) I did not actually think the tonotopy data was required but I guess it does make the case they are examining a lemniscal area. I would be interested in whether the authors feel the results are consistent with those in macaque (PMID 21378972). The gradient running anteromedially looks similar to me and I would not expect much difference (in contrast to the speech).

We thank the reviewer for this suggestion. Also in response to reviewer 2, we now computed the tonotopy for the inferior colliculus (Figure 3—figure supplement 1). We included the following sentence in the manuscript:

“For completeness we ran the same tonotopy analysis also on the inferior colliculi (IC). This analysis (n=28) revealed a single gradient in the IC similarly to previous reports in the macaque (n=3) (Baumann et al., 2011) and human (n=6; n=5) (De Martino et al., 2013; Moerel et al., 2015) (Figure 3—figure supplement 1).”

2) The existence of a significant speech minus non-speech contrast might have fitted a predictive coding account and was clearly predicted by the authors. I found the left vMGB correlation between the [speech vs speaker] contrast and speech recognition performance interesting but rather more nuanced in interpretation and I think that slightly reduces the impact of the work.

We thank the reviewer for this important comment. We hypothesized from the beginning two points, both of which were based on results from previous studies (Díaz et al., 2012; von Kriegstein et al., 2008): (1) a task-dependent modulation exemplified through the speech – speaker contrast, and (2) a positive correlation between the task-dependent modulation and the speech recognition performance. We weighted the importance of these hypotheses equally. We do not think that the main effect is more valuable than the correlation, because the correlation indicates that the task-dependent modulation is behaviorally relevant – a finding that is expected under a predictive-coding account. Important for the present aim of the paper is that the correlation is present in vMGB – something that previous reports on similar correlations (von Kriegstein et al., 2008) were not able to show based on the lack of spatial resolution. Whether the same can be said from a main effect is unclear from the present study. We have now added explanatory sentences in the Introduction and Discussion. In addition, to avoid misunderstanding, we have separated our result section into one that directly describes the results in relation to our two a-priori hypotheses and one section that contains exploratory analyses.

In the Introduction:

“Since our aim of the present paper was to test whether the behaviourally-task dependent modulation of MGB is present in the vMGB, we tested two hypotheses. We hypothesized (i) a higher response to the speech than to the control (speaker) task in the tonotopically organized left and right vMGB, and (ii) a positive correlation between speech recognition performance and the task-dependent modulation for speech in the tonotopically organized left vMGB. Within our design these hypotheses could be addressed by (i) the main effect of task (speech task vs speaker task) in bilateral vMGB and by (ii) a correlation between the contrast speech task vs speaker task with speech recognition performance across participants in left vMGB.”

In the Discussion:

“The results are based on the test of two equally weighted hypotheses: a main effect and a correlation. Although we found no significant main effect in the vMGB (nor in any other subregion of the MGB), the correlation between the task-dependent modulation and behavioral performance was, as hypothesized, significant within the left vMGB.”

Reviewer #2:[…] Major comments: There are less than a handful of studies examining the human auditory thalamus at ultra-high fields-a severe paucity that limits our understanding of thalamic function in processing behaviorally-relevant signals in humans. Within this context, the goals of the present study are laudable. The study convincingly shows tonotopic representation in the vMGB, providing a large-scale replication of Moerel et al., 2015. I'm not at all convinced about the other main thrust of the study related to task modulation/speech recognition. The results show no statistically significant difference in vMGB activity as a function of task. The correlation analyses yield a weak association, but the nature of this association is unclear. What does it mean that modulation in the left vMGB correlates with the speech recognition scores when the task performance in speech vary very little, and there are substantial accuracy differences between the speech and the speaker task? The authors provide three potential explanation for their lack of replication of prior work and seem to suggest that the most likely explanation is that MGB modulation might also have played a role in performing the speaker task…I agree, but this also suggests a critical flaw with the study design which a priori defined task modulation on the basis of a difference in the speech>speaker contrast.

We thank the reviewer for the positive and constructive comments. We address them in three points:

a) effect size of the correlation

The reviewer finds that the correlation analyses show a weak association between task-dependent modulation and speech recognition performance. However, according to Cohen the effect size of the correlation is not weak, but large (Cohen, 1988, p. 80). Cohen describes correlation coefficients (Pearson’s r) in terms of effect sizes and R² values, which are interpreted as the amount of variance explained by the effect. A correlation coefficient of r = 0.46, which Cohen, 1988, deems a large effect in relative terms, results in an R²=0.21. We included the following sentence in the manuscript:

“The correlation coefficient r=0.46 (R²=0.21) is considered a large effect (Cohen, 1988, p. 80) by explaining 21% of the variance of either variable when linearly associated with the variance in the other.”

b) meaning of the correlation

The reviewer asked about the meaning of the correlation (given the difference of 10% between performance on the easier speech task and the more difficult speaker task). The correlation between the task-dependent modulation of the left vMGB and the speech recognition performance across participants means that those participants that displayed a larger task-dependent modulation of left vMGB responses, also achieved a higher amount of hits in the speech task. This indicates that the task-dependent modulation of the left vMGB is behaviourally relevant.

The reviewer asks whether the task performance between 85% and 95% is biologically relevant. We are convinced that this range is relevant. As an example, a 10% difference in task performance when listening to 1000 syllables would result in 100 more syllables correctly identified (correct hits) for those that were better at the task compared to those that were worse at it. We consider this a relevant figure, particularly for such an important aspect of human communication as speech recognition. Previous publications show similar behavioral ranges for speech recognition (for example (Harris et al., 2009; Panouillères and Möttönen, 2018)) where even a 5-10% difference in speech recognition differentiates the hearing impaired from the normal hearing participants. We have now added a clarification in the manuscript:

“The correlation indicated that those participants on the lower side of the task-dependent modulation spectrum, as given by the speech vs speaker contrast, have a lower proportion of hits in the speech task and those participants on the higher side of the task-dependent modulation spectrum, show a higher proportion of hits in the speech task and are thus better at speech recognition. […] However a similar study in the auditory modality is so far missing.”

c) potential flaw in the study design

We do not consider the use of natural speaker voices a potential flaw in the study design since a large influence of dynamic cues in the speaker task was unexpected. We rather consider the null-finding of the present study for the main effect as a potential to initiate further research. Reviewer 3 prompted us to formally assess via a meta-analysis whether the lack of main effect is really a difference between studies or just a merely unlucky result. The results of the meta-analyses imply that indeed there is a difference in the main effect between studies (see response to reviewer 3). Whether this difference in studies is really due to participants using dynamic speaker cues cannot be answered with the present data, but is a testable hypothesis for future studies.

Reviewer #3:[…] 1) Speech > speaker task activation in vMGB – the authors remark that their failure to observe this effect is: "surprising, as this categorical task-effect was observed in three previous experiments in participants with typical development (von Kriegstein et al., 2008, Díaz et al., 2012)." Then in the Discussion section they offer three potential factors that might mediate this surprising outcome.However, all of this discussion is building on a null finding. The absence of evidence for a speech task effect is not evidence of absence. Further analysis is necessary to show which of two interpretations is more likely: (1) that this is an underpowered or merely unlucky study, or (2) a true change from the expected positive to a null (or negative?) effect. Positive evidence of a change in the magnitude of the effect could come from a between-experiment comparison (using activation vs rest as a dependent measure to ensure cross-experiment findings are comparable despite differences in field strength, experimental design, etc.). Or they could use a Bayesian or meta-analytic method to determine whether the expected, opposite or null hypothesis is more likely for the current experiment when combined with a prior from previous studies. Without these analyses, though, the three paragraphs of speculation regarding possible causes of a change in experimental outcome are irrelevant. No compelling statistical evidence for a change has been provided.

We thank the reviewer for this important and helpful comment. We have now conducted meta-analyses (for details see below), and found a net large positive effect size for the task-dependent modulation in MGB over experiments (Figure 8 and Figure 8—figure supplement 1). The sample size of the current experiment is approximately twice as large (n=33) as the other studies included in the meta-analysis (n = [17, 16, 14, 17]), and we hence are convinced that the study was adequately powered. Four studies with overlapping confidence intervals show a positive main effect of task and only the present study falls out of this scheme. We therefore conclude that the difference between the four other studies and the present one in task-dependent modulation is real.

We have now included the analysis in the Materials and methods section:

“Meta-analysis on the main effect of task (speech vs speaker task contrast)

The lack of statistical significance for the speech vs speaker contrast raised the question whether the overall effect is different from the ones reported previously (Diaz et al., 2012; von Kriegstein et al., 2008). […] Effect sizes and standard errors were entered into a random effects model which was estimated with maximum likelihood using JASP 0.9 (jasp-stats.org).”

In the Results section we added:

“We performed a random effects meta-analysis to test whether the (non-significant) effect of the main effect of task in the present study (i.e., speech vs speaker task contrast) was different from other studies that have reported a significant task-dependent MGB modulation for speech. […] We detail potential reasons for this difference in the task-dependent modulation between the studies in the discussion.”

In the Discussion section we added:

“The lack of a significant main effect of task (speech vs speaker) in the vMGB was surprising, and the meta-analyses showed that the null-finding was indeed different from categorical task-effects (speech vs loudness tasks and speech vs speaker tasks) observed in other experiments in participants with typical development (Díaz et al., 2012; Mihai et al., 2019; von Kriegstein et al., 2008).”

2) Correlation with individual differences in speech perception performance – despite the lack of a categorical task effect, the authors use their correlation with behaviour as evidence that: "confirmed our hypothesis that the left first-order thalamic nucleus – vMGB – is involved in speech recognition." Later in the same paragraph they go on to argue that: "The results can be explained neither by differences in stimulus input in the two conditions, as the same stimuli were heard in both tasks, nor by a correlation with general better task performance, as there was no correlation with the speaker task."Here again, they are arguing for a difference between a positive finding (correlation) and a null finding (no correlation) based on the difference between a p-value less than 0.05 and another above 0.05. This is fallacious. They should statistically compare the two correlations to confirm that there is a difference between the speech and speaker findings in order to confirm that some task-specific, auditory aspect of speech perception, rather than generic attentional, executive or motoric factors (that can also be associated with performance) are linked to these regions of the auditory thalamus.

We thank the reviewer for prompting us to reevaluate the control correlation with the speaker task and to compute differences between the correlations. We found a mistake in the previous control correlation analysis, which has now been corrected. The new results do not permit to claim that the behaviorally relevant task-dependent modulation is specific to the speech task and we have toned down our claims on this. Unfortunately, it is impossible to test the specificity for the task-dependent modulation for speech with the present data set. For the visual modality (and the lateral geniculate body, LGN) we have previously found first evidence for the specificity of task-dependent modulation for speech (Diaz et al., 2018). However, such an experiment still needs to be done for the auditory modality. For the current paper, the main goal was to test whether the correlation between task-dependent modulation and speech-task performance is present in vMGB – something that previous reports on similar correlations (von Kriegstein et al., 2008) were not able to show.

We now write on:

“Specificity of the behaviorally relevant task-dependent modulation for speech. In exploratory control analyses we checked whether we could test for a specificity of the correlation between the task-dependent modulation for speech (i.e., the speech task vs speaker task contrast) and the speech recognition behavior across participants. […] The results indicated that although there is a behaviorally relevant task-dependent modulation in the vMGB for speech, we currently do not know whether it is specific to speech recognition abilities.”

Discussion:

“In the present study, the results cannot be explained by differences in stimulus input in the two conditions, as the same stimuli were heard in both tasks. […] However a similar study in the auditory modality is so far missing.”

Additionally, since a lack of behaviourally relevant task-dependent modulation of the other MGB subsection could also be due to just thresholding we now included an exploratory analysis to confirm the specificity of vMGB involvement:

“Specificity of the behaviorally relevant task-dependent modulation to the vMGB. […]Thus this is a first indication that the behaviourally relevant task-dependent modulation for speech is specific to the left vMGB.”

Summary: These two stringent criticisms challenge the authors to do more with the data that they have collected. Nonetheless, I think that the work that these authors have done to localise the tonotopic regions of the auditory thalamus has considerable merit (though it is too far outside my main interest for me to assess this contribution in detail). Yet, I still think further statistical evidence is required for them to associate this thalamus region with a single, specifically-auditory aspects of speech perception (and not speaker perception or generic task abilities). I'd like to see the authors do this additional work in order to more convincingly establish the ventral MGB as part of the neural circuitry for speech perception.

We thank the reviewer for the encouraging remarks and helpful criticisms and believe that addressing them has made our paper stronger. The additional analyses supported our main conclusions and provided the following key points:

· The meta-analyses showed an overall positive effect for the Speech – control task contrast across studies in the MGB,

· the effect of the speech – control task contrast in the current study was different from the other experiments testing the same (or a similar) contrast,

· the correlation between task-dependent modulation and the speech recognition performance across participants in the vMGB was robust against outliers and was present only in vMGB but not in other MGB subsections

· the difference between the two brain-behavior correlations (task-dependent modulation correlated with speech behavioral score and task-dependent modulation correlated with speaker behavioral score) was not significant and we have therefore toned down our claims on speech specificity.

References

Baumann S, Griffiths TD, Sun L, Petkov CI, Thiele A, Rees A. 2011. Orthogonal representation of sound dimensions in the primate midbrain. *Nature Neuroscience* 14:423–425. doi:10.1038/nn.2771

Cohen J. 1988. Statistical Power Analysis for the Behavioral Sciences, 2nd ed. Lawrence Erlbaum Associates. doi:10.1016/C2013-0-10517-X

Davis MH, Johnsrude IS. 2003. Hierarchical Processing in Spoken Language Comprehension. *The Journal of Neuroscience* 23:3423–3431. doi:10.1523/JNEUROSCI.23-08-03423.2003

De Martino F, Moerel M, van de Moortele P-F, Ugurbil K, Goebel R, Yacoub E, Formisano E. 2013. Spatial organization of frequency preference and selectivity in the human inferior colliculus. *Nature Communications* 4:1386. doi:10.1038/ncomms2379

Díaz B, Hintz F, Kiebel SJ, Kriegstein K von. 2012. Dysfunction of the auditory thalamus in developmental dyslexia. *PNAS* 109:13841–13846. doi:10.1073/pnas.1119828109

Gordon-Salant S, Fitzgibbons PJ. 1993. Temporal Factors and Speech Recognition Performance in Young and Elderly Listeners. *Journal of Speech, Language, and Hearing Research* 36:1276–1285. doi:10.1044/jshr.3606.1276

Harris KC, Dubno JR, Keren NI, Ahlstrom JB, Eckert MA. 2009. Speech Recognition in Younger and Older Adults: A Dependency on Low-Level Auditory Cortex. *J Neurosci* 29:6078–6087. doi:10.1523/JNEUROSCI.0412-09.2009

Huth AG, de Heer WA, Griffiths TL, Theunissen FE, Gallant JL. 2016. Natural speech reveals the semantic maps that tile human cerebral cortex. *Nature* 532:453–458. doi:10.1038/nature17637

Mihai PG, Tschentscher N, Kriegstein K von. 2019. The role of the ventral MGB in speech in noise comprehension. *bioRxiv* 646570. doi:10.1101/646570

Moerel M, De Martino F, Uğurbil K, Yacoub E, Formisano E. 2015. Processing of frequency and location in human subcortical auditory structures. *Sci Rep* 5. doi:10.1038/srep17048

Panouillères MTN, Möttönen R. 2018. Decline of auditory-motor speech processing in older adults with hearing loss. *Neurobiology of Aging* 72:89–97. doi:10.1016/j.neurobiolaging.2018.07.013

Price C, Thierry G, Griffiths T. 2005. Speech-specific auditory processing: where is it? *Trends in Cognitive Sciences* 9:271–276. doi:10.1016/j.tics.2005.03.009

Schuirmann DJ. 1987. A comparison of the two one-sided tests procedure and the power approach for assessing the equivalence of average bioavailability. *Journal of pharmacokinetics and biopharmaceutics* 15:657–680.

von Kriegstein K, Patterson RD, Griffiths TD. 2008. Task-Dependent Modulation of Medial Geniculate Body Is Behaviorally Relevant for Speech Recognition. *Current Biology* 18:1855–1859. doi:10.1016/j.cub.2008.10.052